palaeontology

Northwest China, extinctions, Triassic, palaeo-food web, palaeocommunity stability

**Author for correspondence:**
Zhong-Qiang Chen
e-mail: zhong.qiang.chen@cug.edu.cn

# Ecological dynamics of terrestrial and freshwater ecosystems across three mid-Phanerozoic mass extinctions from northwest China

Yuangeng Huang[1,4], Zhong-Qiang Chen[1], Peter D. Roopnarine[1,4], Michael J. Benton[5], Wan Yang[6], Jun Liu[7], Laishi Zhao[2], Zhenhua Li[3] and Zhen Guo[1]

[1]State Key Laboratory of Biogeology and Environmental Geology, [2]State Key Laboratory of Geological Processes and Resource Geology, and [3]School of Computer Science, China University of Geosciences, Wuhan 430074, People's Republic of China
[4]Department of Invertebrate Zoology and Geology, California Academy of Sciences, San Francisco, CA 94118, USA
[5]School of Earth Sciences, Life Sciences Building, Tyndall Avenue, University of Bristol, Bristol BS8 1TQ, UK
[6]Geology and Geophysics Program, Missouri University of Science and Technology, Rolla, MO 65409, USA
[7]Key Laboratory of Vertebrate Evolution and Human Origins of Chinese Academy of Sciences, Institute of Vertebrate Paleontology and Paleoanthropology, Chinese Academy of Sciences, Beijing 100044, People's Republic of China

Z-QC, 0000-0001-5341-6913; MJB, 0000-0002-4323-1824

The Earth has been beset by many crises during its history, and yet comparing the ecological impacts of these mass extinctions has been difficult. Key questions concern the kinds of species that go extinct and survive, how communities rebuild in the post-extinction recovery phase, and especially how the scaling of events affects these processes. Here, we explore ecological impacts of terrestrial and freshwater ecosystems in three mass extinctions through the mid-Phanerozoic, a span of 121 million years (295–174 Ma). This critical duration encompasses the largest mass extinction of all time, the Permian–Triassic (P–Tr) and is flanked by two smaller crises, the Guadalupian–Lopingian (G–L) and Triassic–Jurassic (T–J) mass extinctions. Palaeocommunity dynamics modelling of 14 terrestrial and freshwater communities through a long sedimentary succession from the lower Permian to the lower Jurassic in northern Xinjiang, northwest China, shows that the P–Tr mass extinction differed from the other two in two ways: (i) ecological recovery from this extinction was prolonged and the three post-extinction communities in the Early Triassic showed low stability and highly variable and unpredictable responses to perturbation primarily following the huge losses of species, guilds and trophic space; and (ii) the G–L and T–J extinctions were each preceded by low-stability communities, but post-extinction recovery was rapid. Our results confirm the uniqueness of the P–Tr mass extinction and shed light on the trophic structure and ecological dynamics of terrestrial and freshwater ecosystems across the three mid-Phanerozoic extinctions, and how complex communities respond to environmental stress and how communities recovered after the crisis. Comparisons with the coeval communities from the Karoo Basin, South Africa show that geographically and compositionally different communities of terrestrial ecosystems were affected in much the same way by the P–Tr extinction.

## 1. Introduction

There have been many mass extinctions in the history of life, and these all seem to have had unique features. However, mass extinctions provide a series of

natural experiments of varying levels of severity from which biologists can seek common features of the killing mechanisms and the subsequent recovery. A series of crises from the early Permian to Early Jurassic, some 295–174 million years ago (Ma), may have shared a common physical driver that relates to the current environmental crisis, namely extinction following the massive release of carbon dioxide and other greenhouse gases into the atmosphere [1,2]. The evolving large igneous province killing model includes sharp global warming, acid rain, mass wasting, ocean acidification and stagnation [3].

Three of the mid-Phanerozoic mass extinctions, the Permian Guadalupian–Lopingian (G–L), Permian–Triassic (P–Tr) and Triassic–Jurassic (T–J), are of differing magnitude, with 64–80% [4,5], approximately 89% and approximately 41% [6] losses of terrestrial tetrapod genera, respectively. And yet global conditions (continental configurations, large igneous province eruptions, climates) were similar. These extinction events laid the foundation of our modern marine and terrestrial biosphere, but the community dynamics and ecosystem responses to extreme perturbations during these critical periods still remain poorly understood. And given ongoing anthropogenic changes to the ecosystem, it has become necessary to understand the properties of communities as they approach the extremes of their stable ranges [7].

Most studies of mass extinction ecology and dynamics have focused on marine environments where fossil records are assumed to be more uniformly sampled and hence more comparable than terrestrial fossil records. However, the terrestrial environment cannot be ignored: today there are many more species on land than in the oceans, although that was likely not the case in the Permian to Jurassic, but the effects of catastrophic environmental change might be expected to be different, perhaps even more severe on land than in the oceans, which can show some buffering against atmospheric changes [2,8]. Further, the ecological methods we deploy here require excellent spot sampling but not excellent continuous fossil records. In other words, the methods work with 'snapshots' of life, individual fossil assemblages that document life reasonably completely at one place during one interval time. Stabilizing the sampling to a single geographic region, here Xinjiang in China, provides some continuity and comparability of biotic assemblages that occupied similar palaeolatitudes through a long span of time.

Successive analyses of trophic structure over deep time can be employed to assess important ecological questions. Recently developed food web models are a powerful tool for exploring and quantifying community dynamics, simulating how the structures of modern [9–12] and ancient [7,13–17] communities influence their responses to perturbations. Therefore, we provide results from cascading extinction on graphs (CEG) modelling [13] of terrestrial ecosystems to quantify community resistance during the G–L, P–Tr and T–J transitions. Furthermore, we simulate interspecific trophic interactions and community dynamics to determine the effect, if any, of taxonomic and ecological extinctions on community structure and stability. The responses of communities to disturbance, in terms of declining productivity, are used to model the resistance of the communities to perturbation, and then quantify the ecological dynamics of communities before and after these extinctions [7].

# 2. Material and methods

## (a) Geological setting and palaeoclimate

During the Permian to Jurassic, the Junggar basin of modern Xinjiang, China was located on the Junggar block, a part of the Pangaea supercontinent (electronic supplementary material, figure S1a). According to palaeogeographic reconstructions, this basin was located at a latitude of about 45° N from the Permian and during the entire Mesozoic [18]. The chronostratigraphy of the Junggar basin is constrained mainly by biostratigraphy of invertebrates and volcanic ash radioisotopic ages [19–21]. The early Permian to Early Jurassic deposits include 12 formations (electronic supplementary material, figure S1d) that mainly correspond to alluvial and lacustrine settings [19]. The Sakamarian climate was highly variable subhumid-semiarid, and great climatic variability persisted during the Artinskian and, perhaps, Kungurian. A prominent climatic shift from highly variable subhumid-semiarid to humid–subhumid conditions occurred across the Artinskian-Capitanian unconformity. Stable humid–subhumid climate dominated from the Capitanian to early Induan. A clear shift to highly variable subhumid-semiarid conditions occurred in the middle Induan and persisted to the end of the Olenekian [19]. Subhumid conditions returned during the Ladinian, then subhumid-humid conditions in the Carnian [19,22]. The Norian climate was warm and humid and was followed by an increase in temperature and humidity during the Rhaetian to Sinemurian. Finally, warm and dry climatic conditions returned in the Pliensbachian and Toarcian [23].

## (b) Database

The complete data used in this study are stored in the Dryad Digital Repository. In summary, we conducted an in-depth literature review to maximize the completeness and robustness of our early Permian to Early Jurassic dataset for nonmarine species in northern Xinjiang, China. Based on updated stratigraphic information and taxonomic revisions, a database of 436 species among 10 major clades (reptiles, synapsids, amphibians, fish, notostracans, insects, bivalve molluscs, gastropods, conchostracans and ostracods) was constructed at the formation/member level (electronic supplementary material, table S1). The insect body fossil record of north Xinjiang is limited in certain formations; here we used the approach of [14] to estimate insect richness in communities with poorly preserved insect fauna. This approach uses a linear relationship ($S_i = 45.267r + 46.997$, where $S_i$ is the predicted richness of guild i and r is the link ratio) between insect species richness, and the ratio of insectivore richness to the number of insectivore prey guilds, recognizing that most of the insectivores belong to insectivore-carnivore guilds [24]. The estimated insect richness of each community was partitioned among herbivorous, omnivorous and carnivorous guilds based on observed ratios from the late Permian *Daptocephalus* Assemblage Zone and Middle Triassic *Cynognathus* Asssemblage Zone of the Karoo Basin, South Africa [7]. Sampling and uneven strata issues are discussed in electronic supplementary material, section S1.

Here, a total of 14 palaeocommunities spanning approximately 121 million years from the early Permian to Early Jurassic was assembled based on the fossil records from multiple sections in north Xinjiang (electronic supplementary material, figure S1c), namely the Permian Lucaogou (LCG; ~Roadian), Hongyanchi (HYC; ~Wordian), Quanzijie (QZJ; ~Capitanian), Wutonggou (WTG; ~Wuchiapingian), Lower Guodikeng (LGDK; ~Changhsingian) formations, the Triassic Upper Guodikeng (UGDK; ~Griesbachian), Jiucaiyuan (JCY; ~Dienerian–Smithian), Shaofanggou (SFG; ~late Olenekian), Lower Kelamayi (LKLMY; ~late Anisian–Ladinian), Upper Kelamayi (UKLMY; ~Carnian), Huangshanjie (HSJ; ~Norian)

Proc. R. Soc. B 288: 20210148

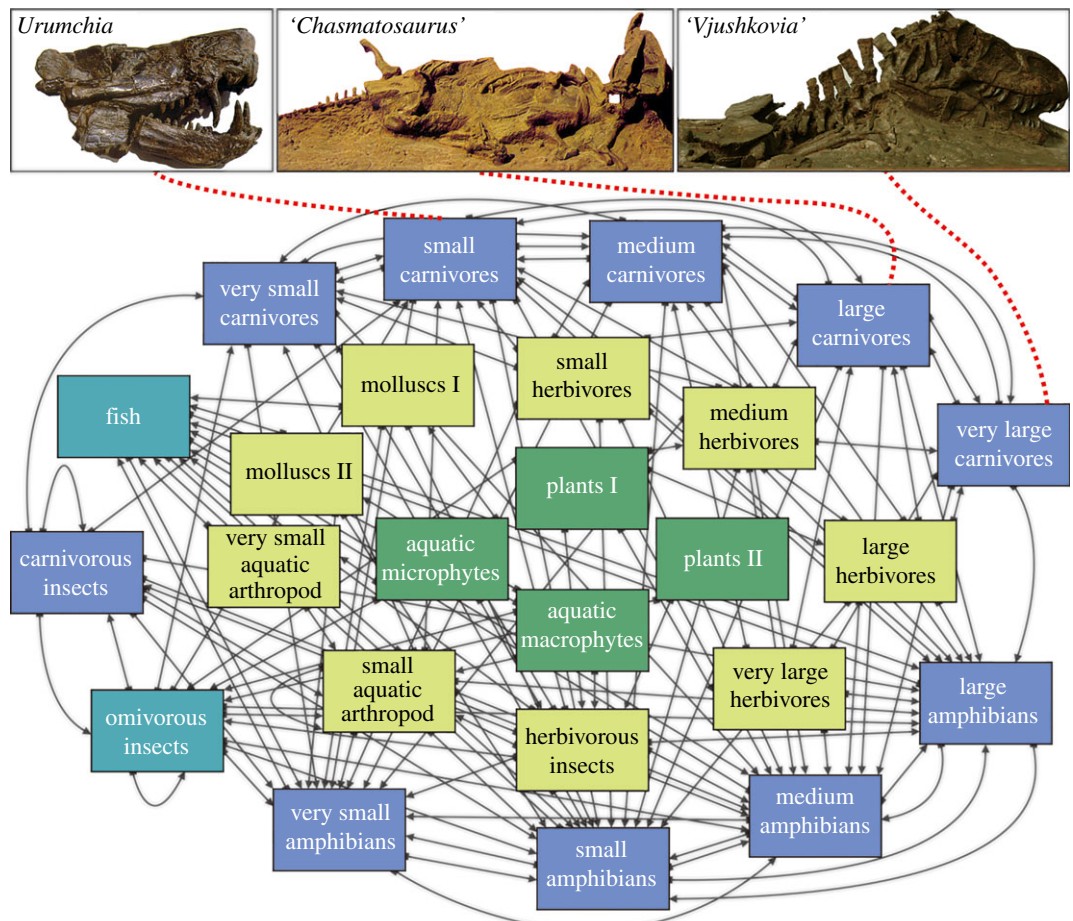

**Figure 1.** Schematic of metanetwork applicable to all the palaeocommunities tested here, with representative predators. Guilds are shown in boxes, and arrows indicate the direction of trophic interaction, pointing from prey to predators. Each guild comprises species that would have interacted with at least some species out of the set of guilds to which their guild is linked. (Online version in colour.)

and Haojiagou (HJG; ~Rhaetian) formations, and the Lower Jurassic Badaowan (BDW; ~Hettangian–Sinemurian) and Sangonghe (SGH; ~Pliensbachian–Toarcian) formations (electronic supplementary material, figure S1d).

## (c) Palaeocommunity reconstruction

Here, we define a guild as a set of species whose members share ecologically relevant characteristics, namely habitat, body size ranges and potentially the same predator and prey when present in the same community (e.g. carnivore, herbivore) [25,26]. Therefore, we assigned all taxa to a series of guilds based on their ecological functional types (electronic supplementary material, table S2). Trophic links between guilds were inferred using extensive literature surveys of species life mode, feeding habits, functional morphology (e.g. feeding apparatus), habitat, species associations and living analogue species [26,27], to make a metanetwork (guild-level food web; electronic supplementary material, table S3). Reptiles and synapsids were divided into ten guilds based on their diet and size, namely very small (skull length 0–100 mm), small (101–200 mm), medium (201–300 mm), large (301–400 mm) and very large (401 mm and above). We defined herbivores as feeding on a single producer guild, whereas carnivores preyed on amniote herbivores and carnivores up to two size classes larger and smaller than themselves, with the smallest two carnivore guilds also preying on arthropods. Amphibian guilds preyed on amphibian and amniote guilds up to two size classes larger and smaller than themselves, as well as fishes, insects and aquatic invertebrates. Fish fed on aquatic producers, insects, aquatic invertebrates and temnospondyls. Insects were divided into herbivorous, omnivorous and predatory guilds. Non-insect invertebrate

guilds include molluscs I (bivalves), molluscs II (gastropods), very small aquatic arthropods (conchostracans and ostracods) and small aquatic arthropods (notostracans) (figure 1).

There is a finite ensemble of species-level food webs consistent with the assigned links between guilds. The species-level food webs vary in the distribution of interspecific trophic links, which we stochastically generated by applying mixed exponential–power-law link distributions uniformly to all 14 communities, consistent with the hyperbolic distributions typical of modern food webs [7,26] and other complex networks [28]. This method of stochastically assigning links using an empirically derived power-law distribution allows our model to capture the structure and spatiotemporal variations of a real food web in the ensemble. In general, this variation does not generate significant differences of dynamics among food webs in the ensemble, being constrained and dictated by the higher level guild organization and interaction of the community [14,29,30].

## (d) Functional composition and trophic space

Dissimilarity among the palaeocommunities based on their functional structures was analysed in a two-step procedure. First, palaeocommunities were ordinated according to the presence or absence of guilds using nonmetric multidimensional scaling (NMDS) based on intercommunity Jaccard distances. The Jaccard distance accounts for overlap in guild composition only, ignoring taxon richness within guilds. Second, the NMDS analysis was repeated, but this time using Bray–Curtis dissimilarity indices, with both the presence of guilds and their taxon richness accounted for [7]. Here, we use trophic space to describe the metanetwork range of the community. Jaccard distances were applied in NMDS analyses for the metanetwork plus its transposed

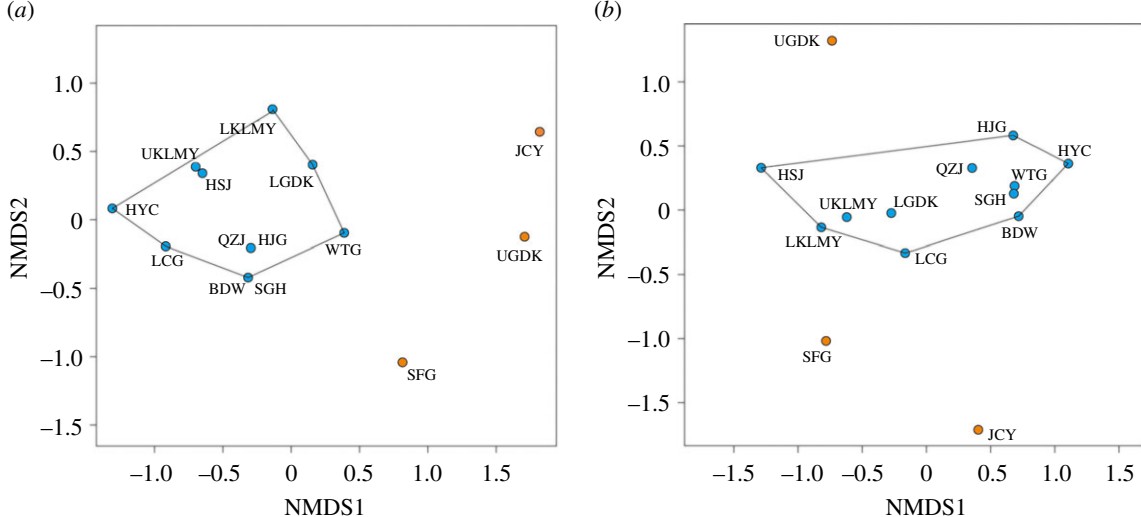

**Figure 2.** NMDS ordinations of palaeocommunities using Jaccard (*a*) and Bray–Curtis (*b*) distances. Early Triassic palaeocommunities include UGDK, JCY and SFG. (Online version in colour.)

## (e) Modelling of palaeocommunity dynamics

Here we assess stability and palaeocommunity dynamics using the cascading extinction on graphs (CEG) model, designed specifically to take into account uncertainty in palaeontological community data [25]. Even a modern food web reconstruction can never capture all species interactions, let alone a palaeo-food web. Therefore, CEG stochastically generates 100 species-level food webs by randomly and repeatedly drawing species from the trophic guilds and assigning them as predators and prey within the constraints of the metanetwork assembled, as described earlier. Each species-level food web is a hypothetical representation of the palaeocommunity, and each species is constrained to the potential interactions for its guild, as specified by the metanetwork. For each species-level food web, we simulated bottom-up disturbance by incrementally reducing primary productivity and recorded the resulting secondary extinction (the proportion of consumers that became extinct at a given perturbation level). Unlike many food web models, extinction is not purely topological; CEG permits top-down interactions and trophic cascades resulting from bottom-up perturbations by recalculating interaction strengths of species when some of their resources or predators go extinct. Primary productivity was modelled as a function of the density of herbivorous interactions [13], scaling productivity as ten times that of herbivore richness in accordance with general assimilation efficiency between trophic levels (e.g. [14,15]).

Differences between pre- and post-extinction stability and resistance would also suggest that trophic restructuring occurred, altering palaeocommunity stability. The CEG model was applied to examine the resistance or vulnerability of the palaeocommunities to secondary extinction when one or more components of the communities were perturbed. In this case, primary production was the quantity perturbed, and vulnerability was measured as the fraction of species that became secondarily extinct as a consequence. In general, this type of CEG perturbation yields a response where a community's resistance is high, and relatively uniform over a broad range of perturbations, as the magnitude of perturbation is increased incrementally [7]. Community composition is thus stable over this range of variation in primary productivity.

In almost all cases examined, however, a point of perturbation is reached where secondary extinction increases dramatically. This is defined as a collapse threshold, and reductions of primary production beyond this threshold are expected to change the community significantly, reducing richness and altering the composition. We therefore quantified community resistance, or compositional stability, based on variations of collapse thresholds (perturbations that resulted in the largest, often abrupt, increase in secondary extinction, identified by changepoint analysis), with higher collapse thresholds indicating higher resistance or compositional stability. Principal components analysis was also used to capture the CEG dynamics at relatively low perturbation level (electronic supplementary material, section S2). For each community, we calculated the thresholds on all 100 simulations using the R package 'changepoint.np', in which statistical changes in the data sequence were detected by implementing the pruned exact linear time, or PELT, algorithm [31].

This combined approach (i.e. NMDS, network metrics, CEG results) allows us to describe how the structures of the communities differ and then determine how those differences translate into community performance in the face of perturbations. In addition, the structure of species-level food webs was compared between communities using network properties (S, number of taxa; L, number of trophic links; L/S, linkage density; C (L/S$^2$), directed connectance). Network structure properties were calculated using custom code written by P.D.R. in Julia [32].

## 3. Results

### (a) Functional structural and trophic space variations

A total of 25 guilds were recognized from these 14 palaeocommunities, although not all guilds are present in each palaeocommunity (figure 1; electronic supplementary material, table S2). The set of guilds includes four primary producer guilds, and both terrestrial and freshwater invertebrates and vertebrates. A total of 170 inter-guild links were also established (figure 1), defining pairs of guilds where species within one guild prey upon some or all of the species in the other guild.

Pairwise palaeocommunity Jaccard distances and Bray–Curtis dissimilarities were applied in NMDS analyses to ordinate and compare palaeocommunities in terms of their guild compositions and richnesses among the Permian–Jurassic communities (figure 1). The Jaccard distance analysis (figure 2*a*)

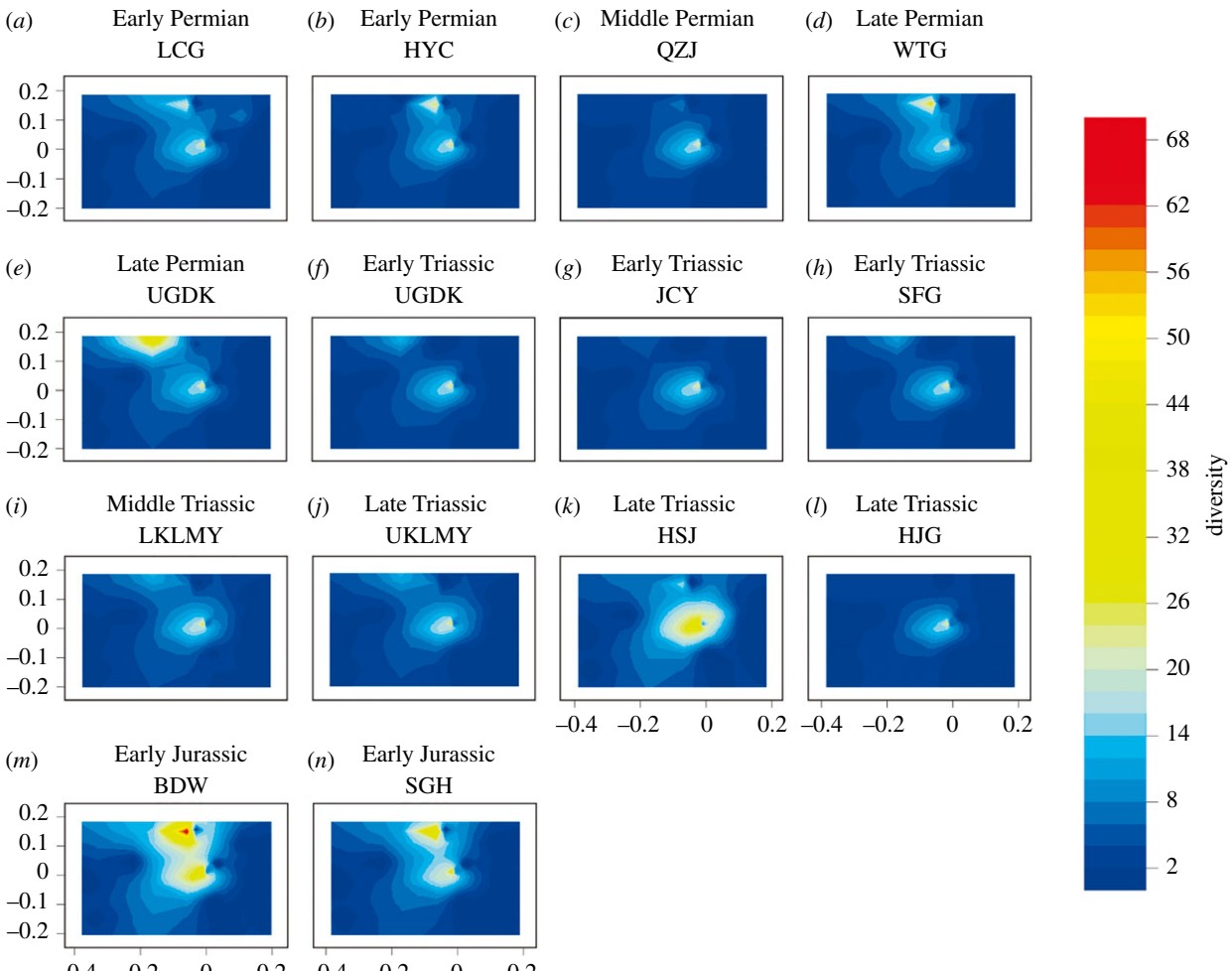

**Figure 3.** Trophic space variations of 14 communities from Xinjiang, with communities arranged in order from old to new. Guilds were ordinated by NMDS analyses, and guild richness is drawn as a contour map. (Online version in colour.)

shows that the three Early Triassic communities (UGDK, JCY, SFG) are distinguished from a main cluster that includes all the other communities, indicating compositional differences among the two groups of communities.

Trophic space analyses show that early Permian to Early Jurassic communities from Xinjiang fall into two main clusters, the upper being the aquatic invertebrates cluster and the lower the insects cluster (figure 3), and trophic space variations are primarily controlled by tehe expansion and shrinking of these two clusters. The trophic space largely shrank in the extinction interval communities (the Capitanian QZJ, Induan UGDK and JCY, and latest Triassic HJG) when compared with the pre-extinction and post-extinction communities (figure 3). No fundamental trophic space innovation or loss was detected for the local Xinjiang communities during this approximately 121 Myr, as such innovation would be manifested on the trophic space plots as the rise of a new peak on the landscape.

## (b) Species-level food web modelling

Connectance (the proportion of possible links that are realized, $L/S^2$) increases in the Capitanian QZJ, Early Triassic UGDK, JCY and SFG, and latest Triassic HJG community, and decreases in the subsequent recovery periods (namely late Permian, Middle Triassic and Early Jurassic). Mean $L/S$ values maintain low volatility except for a striking spike in the Late Triassic HSJ community (figure 4). Connectance is significantly correlated with diversity (Pearson's correlation

coefficient = 0.85, $p < 0.05$), but not with $L/S$ (Pearson's correlation coefficient = 0.48, $p = 0.0764$).

Pronounced anomalies in community resistance reflected by the CEG results occurred across the G–L, P–Tr and T–J mass extinctions (electronic supplementary material, figures S8, 11–13, 17). Moreover, the Early Triassic JCY and Late Triassic HJG communities have relatively high secondary extinction even at relatively low levels of perturbation (0.1–0.4) (electronic supplementary material, figure S5). Collapse thresholds varied throughout the series, and there were several transitions where thresholds differed significantly between successive communities (ANOVA, $F = 53.75$, $p \ll 0.001$). Threshold values declined stepwise during the middle Permian, and were significantly lower in the Capitanian QZJ community compared to the preceding HYC community, prior to the G–L mass extinction (figure 5b). Thresholds then rebounded rapidly in late Permian. The most significant transition is highlighted by a dramatic and significant decline in threshold value across the P–Tr boundary, with average collapse thresholds decreasing from 0.633 in the LGDK to 0.569 in the UGDK communities (figure 5b). The other two Early Triassic communities (JCY, SFG) also exhibit similar CEG dynamic patterns to the UGDK and have very low average collapse thresholds and highly variable threshold ranges (figure 5b). Thus, community stability and resistance in the earliest Triassic remained low, but highly variable and unpredictable.

Collapse thresholds did not recover to pre-extinction levels until the Middle Triassic, increasing significantly

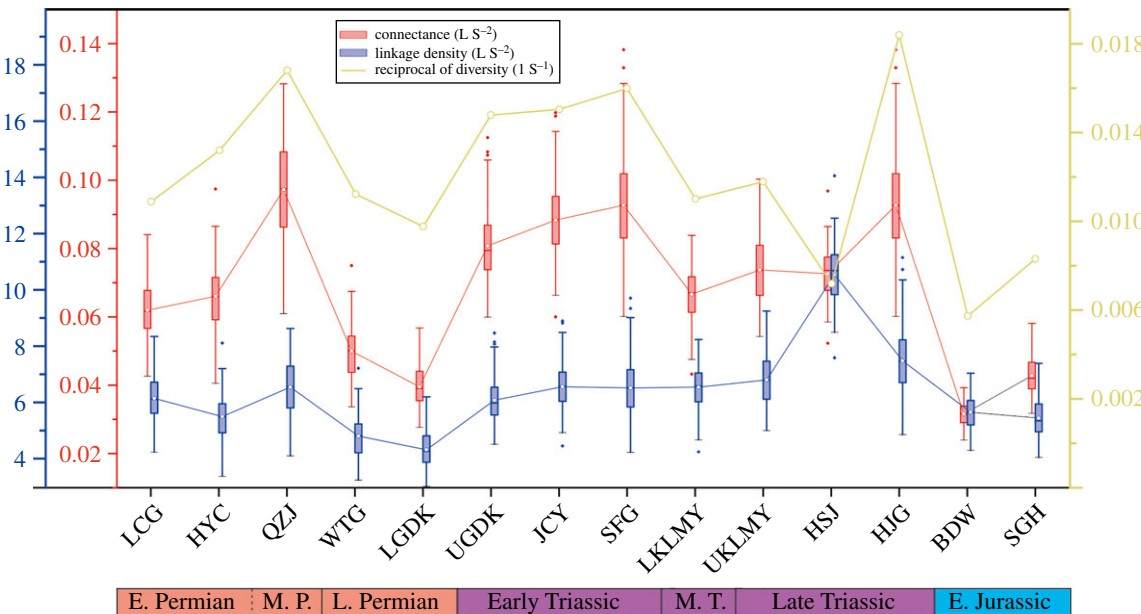

**Figure 4.** Variation in directed connectance ($L/S^2$), L/S (linkage density) and 1/S (reciprocal of species richness) of 14 communities from Xinjiang, ranging from early Permian to Early Jurassic in age. (Online version in colour.)

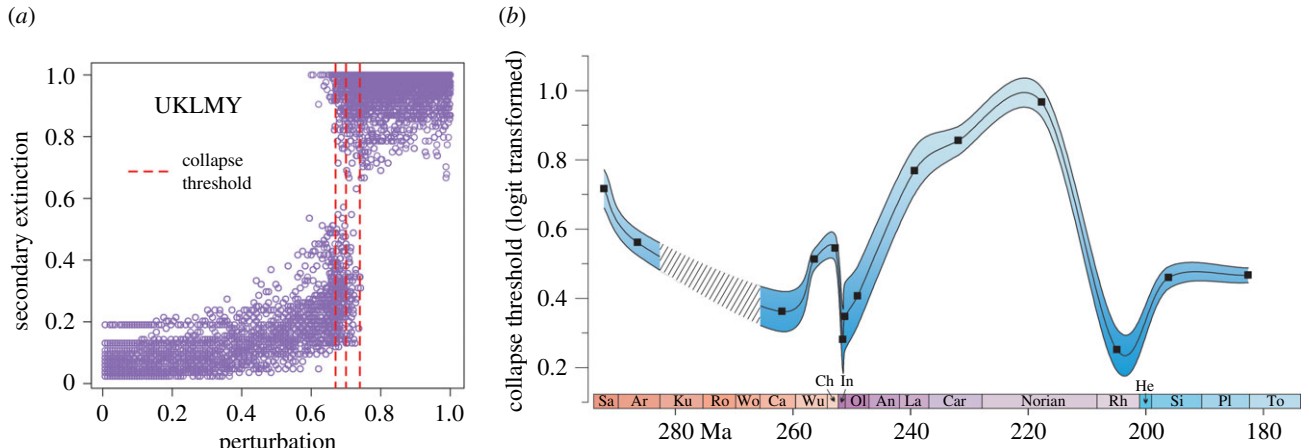

**Figure 5.** Primary productivity perturbation versus secondary extinction for one palaeocommunity and collapse threshold variations of 14 palaeocommunities from middle Permian to Lower Jurassic in northern Xinjiang. (a) Primary productivity perturbation versus secondary extinction for 100 species-level networks generated for the UKLMY, with the collapse threshold marked by the red line. (b) Modified Bezier curves with ± standard errors as blue shading of collapse thresholds (logit transformed). Abbreviations: Sa., Sakmarian; Ar., Artinskian; Ku., Kungurian; Ro., Roadian; Wo., Wordian; Cap., Capitanian; Wu., Wuchiapingian; Ch., Changhsingian; In., Induan; Ol., Olenekian; An., Anisian; La., Ladinian; Car., Carnian; Rh., Rhaetian; He., Hettangian; Si., Sinemurian; Pl., Pliensbachian; To., Toarcian. (Online version in colour.)

between the SFG and LKMY communities, and reaching an acme in the Late Triassic HSJ community. The Middle–Late Triassic communities (LKLMY, UKLMY, HSJ) possessed high, stable thresholds. A sharp decrease in thresholds occurred again in the latest Triassic between the HSJ and HJG communities, then resurged significantly and rapidly in the earliest Jurassic BDW community and remained at relatively high levels in the Early Jurassic (figure 5b).

## 4. Discussion

### (a) Community structure and dynamics over the Permian–Triassic transition

Despite uncertainties about dating the PTB in continental successions, we can be confident we are comparing pre- and post-extinction communities. This is because for studies of this kind, we simply have to be sure that we are looking at communities

below and above the line, even though we cannot perhaps always mark the exact P–Tr mass extinction and PTB horizons in the rock successions. Here, the mass extinction is calibrated at the middle of the Guodikeng Formation based on biotic changeover and organic carbon isotopic correlations with the marine Meishan GSSP, South China [33–37]. Our comparison of the LGDK and UGDK communities shows the major diversity drop, changes in guild richnesses and CEG dynamics expected for the mass extinction from comparisons with South Africa and Russia. The guild compositions of the Early Triassic communities are characterized by the loss of several lineages (e.g. the Bivalvia, Gastropoda), which are the bases for their distinction from the rest of the communities.

Our study does not conform with some recent analyses of marine ecosystems where massive biodiversity loss was detected through the P–Tr mass extinction, as expected, but functional richness remained unaltered [38]. Here, we did detect a significant loss of guilds during the P–Tr transition, although it did not alter the original trophic space structure

(figure 4). This might reflect the relatively high functional redundancy of the terrestrial and freshwater communities in Xinjiang, and the high functional similarity between the new guilds that emerged in the Early Triassic compared to previous guilds. In other words, in marine and terrestrial situations, if functional guilds apparently survive unaffected, or the overall trophic space structure does not collapse, it is important to look more closely to determine whether there truly was continuity of lineages through the crisis, or whether new surviving groups took over functional guilds after their previous occupants had vacated them. In Xinjiang, the P–Tr transition witnessed depleted guild richnesses (with reduced taxon richness in each invertebrate guild), the complete loss of the mollusc guilds, and the appearance of new guilds in the earliest Triassic (very small carnivore, medium carnivore, large carnivore, and small amphibian); this significant decline in guild richnesses coincides with the P–Tr extinction.

Plunging community diversity caused connectance values in the earliest Triassic food webs to increase sharply, from a pre-extinction value of 0.04 (LGDK) to a post-extinction value of 0.08 (UGDK), with continued increases in the Early Triassic (figure 5). It is important to understand that, whereas such connectance values can indicate increasing community complexity if species richness remains constant, here the L/S values increase only slightly. Therefore, the higher connectance values arise from species diversity decrease, and small communities inherently tend to have a high connectance value [39].

The widespread decline in primary production across the terrestrial P–Tr boundary has already been documented empirically [40–43]. Floras (60 species in 32 genera) in Xinjiang also suffered a dramatic drop in biodiversity across the P–Tr extinction horizon, with only seven species in six genera recorded in the Early Triassic [44]. Lycopods and fern-dominated herblands replaced gymnosperm-dominated forests [43,45,46], indicating a reduction of primary productivity, or at least a major change in the diversity of producers. The observed changes in animal diversity also suggest a decline in the amount of productivity that is needed to support the communities, and that in turn is perhaps consistent with the fact that palaeobotanical evidence suggests a drop in plant diversity at this time. CEG dynamics revealed a sharp decrease in thresholds during the P–Tr transition (figure 5) and remained at a low level throughout the Early Triassic, indicating that the possible disruptions in primary productivity in the Xinjiang palaeocommunities in the early Triassic, led to increased sensitivities and reduced resistance of the CEG food web models. This finding is congruent with CEG model results for the P–Tr transition in the Karoo Basin of South Africa, where a similar implied drop in productivity was observed [13]. The Early Triassic communities therefore were less stable than pre-extinction communities, which was probably caused by the loss of species richness, loss of guilds, shrinkage of trophic space, or some combination thereof. Here we cannot point to an obvious factor as the cause, because the nonlinear dynamics of the CEG model make such predictions very uncertain. The P–Tr mass extinction destroyed the community stability of terrestrial and freshwater ecosystems worldwide. This is shown by the comparison of P–Tr communities from southern high latitudes (50–60° south; Karoo basin) and northern moderate latitudes (40–50° north, Xinjiang) (electronic supplementary material, figure S1a), which all show much reduced community stability in CEG dynamics across the P–Tr boundary, coinciding with

biodiversity extinction. Furthermore, the very weak resilience to ecological perturbations and unusual functional structures of Early Triassic communities in the Karoo Basin delayed biotic recovery following the P–Tr mass extinction [47].

Connectance decreased in the Middle Triassic but did not recover to late Permian level, L/S values remained steady throughout the Triassic except for a spike in the HSJ community, which is because of the evolution of highly connected species (e.g. insects and notostracans). However, connectance of the HSJ community did not change much, because increased species richness offset the influence of increased L/S. Threshold values achieved pre-extinction levels in the Middle Triassic LKLMY community, representing the full recovery of community stability after the P–Tr extinction (figure 5; electronic supplementary material, figure S5), which coincides with the final recovery of marine ecosystems [48], and possibly the Middle Triassic recovery of the Karoo Basin ecosystem [47] and the lacustrine ecosystems in the Ordos Basin, North China [49].

Whether the Triassic recovery in various latitudinal regions and compositionally different communities followed the same broad patterns has been discussed, the pattern of ecological recovery of the communities in different marine and terrestrial regimes proceeded in a similar way, despite the different identities of the taxa involved, corroborating the hypothesis that there are taxon-independent norms of community assembly [7,48]. Our results show that the community stability collapse and recovery trajectory of the Karoo and Xinjiang are very similar to one another, although the Karoo communities are dominated by terrestrial taxa (i.e. tetrapods [7,50]), whereas the Xinjiang communities are dominated by aquatic invertebrates. Indeed, those different kinds of communities show many of the same kinds of changes through the Permian and Triassic. This implies that different types of communities were affected in much the same way by the P–Tr mass extinction or that when very different sampling regimes are applied to more or less the same underlying terrestrial community they can recover a consistent signal of ecosystem change in the critical periods.

The significant increase in community stability from the Early to Middle Triassic in Xinjiang probably stems from the emergence of a new guild: small aquatic arthropods, namely the Notostraca. This guild is highly connected, which links with 12 other guilds (figure 1; electronic supplementary material, table S3). High connectivity, or the emergence of a high connectivity guild, might help to stabilize the community in the model because of changes in network dynamics [51]. Also, the generalist trophic habits for this guild would tend to make it less likely to be fatally impacted, because they have more food sources they could draw upon when the community was disturbed. The elevated guild richnesses of other highly connected guilds, such as insects, fishes and amphibians also contributed to some extent to the high stability of Middle and Late Triassic communities.

## (b) Contrasting ecological dynamics during the Guadalupian–Lopingian and Triassic–Jurassic transitions

Community composition showed no significant variations during the G–L and T–J transitions (figure 2), although there was a decline in species richness. Similar to the P–Tr

transition, increase in connectance, trophic space shrinking and decrease in collapse thresholds are observed during the G–L and T–J transitions (figures 3–5). Generalities in the network structure of trophic interactions have been identified for extant and some ancient food webs [16,52,53], and here, our results highlight that generalities may not only exist in ancient food web structure but also in their responses to perturbation during extinction events.

Decreased threshold values also mark major community destabilizations in both the G–L and T–J transitions (figure 5). However, unlike the P–Tr event, where the lowest threshold values were observed in the community immediately after the extinction, values after both the G–L and T–J mass extinctions were lowest in communities prior to the crises. For instance, the Capitanian QZJ community yields the lowest values after a stepwise decline through the early-middle Permian (figure 5). Thus, it is likely that the most vulnerable Permian community might have resulted from a stepwise biodiversity decrease, a pattern observed in both marine and nonmarine fossil records [4,5,54].

A sharp drop in thresholds is also observed in the terminal Triassic (Rhaetian) HJG rather than in the earliest Jurassic BDW community (figure 5). This weakened community resistance unambiguously stems from the overall decrease in the number of guilds and guild richnesses (electronic supplementary material, table S2). The exact T–J extinction horizon has not been well calibrated in North Xinjiang due to the limited fossil record, but our result shows that community stability had already decreased substantially in the Rhaetian, and it is not coincident with the major extinction just below the ammonoid-defined marine T–J boundary [55]. On land, a comprehensive review of nonmarine records concluded that T–J biotic changes were non-uniform (different changes in different places), and long-term changes in the terrestrial biota across the TJB are complex and diachronous [56]. Thus, we argue that a local disruption of ecosystems happened in the Rhaetian in Xinjiang.

Comparable extinction patterns of communities are also observed in the terrestrial Late Cretaceous terrestrial communities in North America [14], where pre-extinction Maastrichtian communities possessed much lower threshold values than Campanian communities, suggesting a more vulnerable community, which may have exacerbated the impact and severity of the end-Cretaceous extinction [14]. Similarly, the vulnerable QZJ and HJG communities may also have facilitated the G–L and T–J mass extinctions, respectively. Collapse thresholds surged again immediately after the G–L and T–J mass extinctions (figure 5), implying a rapid recovery in community stability in both cases. Additional studies of community dynamics from other locations and time periods are necessary to assess the pervasiveness of potential generalities/disparities.

All three of the mass extinctions investigated here may have been driven primarily by large igneous province volcanism and consequent impacts on the environment (sharp warming, acid rain, ocean acidification and stagnation). Therefore, the contrast between the huge P–Tr and the smaller G–L and T–J crises may reflect their relative extinction magnitudes [57], but the rates of environmental changes and differences in environmental effects could also create such a contrast, and further studies are needed to assess these suggestions. The P–Tr extinction damaged ecosystems more severely than the other crises during the Palaeozoic–Mesozoic transition. The differences among these extinction recovery dynamics are consistent with a recent hypothesis claiming that ecosystem compositional and dynamic stabilities are contingent upon histories of functional coevolution among clades, and that only extinctions of magnitudes sufficient to remove guilds are likely to result in post-extinction ecosystem replacement or re-organization [47].

## 5. Conclusion

Our results indicate three significant decreases in palaeocommunity stability in Xinjiang, coinciding with the G–L, P–Tr and T–J extinctions. The G–L and T–J transitions were each preceded by low-stability communities, but the subsequent recoveries were rapid. However, ecological recovery from the P–Tr mass extinction was prolonged, and the Early Triassic terrestrial and freshwater communities showed low stability and highly variable and unpredictable responses to perturbation primarily due to the huge losses of species, guilds. We confirm that the unusually low community stability in the Early Triassic, which was first observed in the Karoo Basin, is a global phenomenon. Increased connectance, reduced trophic space and decreased collapse thresholds are observed in all three extinction events, suggesting that generalities not only exist in ancient food web structure but also in their responses to perturbation during extinction events. Global correlations show that the same extinction and recovery patterns of terrestrial ecosystems exist not only across a wide geographic range, but also among compositionally different communities

Data accessibility. The datasets supporting this article have been uploaded as part of the electronic supplementary material.

Authors' contributions. Y.H., L.Z., Z.L., Z.G. W.Y. and J.L. compiled the data; Y.H. and P.D.R. carried out analyses; Y.H., Z-Q.C., P.D.R. and M.J.B. led the writing; all co-authors provided ideas and various contributions.

Competing interests. The authors declare no competing interests.

Funding. Financial support was provided by the National Natural Science Foundation of China, NSFC grant nos. 41821001 and 4193000537 (to Z.-Q.C.), 41902315 (to Y.H.), 92055212 (to L.Z.), the United States National Science Foundation, NSF grant EAR 1714898 (to P.D.R.), the Subject Construction funds (CUG grant no. 162301192694), and the State Key Laboratory of Biogeology and Environmental Geology (GBL11905).

Acknowledgements. We are grateful for the insightful reviews from J. Botha-Brink, N. Brocklehurst, two anonymous referees and the editorial board that greatly improved the manuscript. This paper is a contribution to Center for Geoscience Knowledge Graph of China University of Geosciences.

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
