## [Peer Review File · Proceedings of the Royal Society B: Biological Sciences]

Review History

RSPB-2020-2404.R0 (Original submission)

Review form: Reviewer 1

Recommendation

Major revision is needed (please make suggestions in comments)

Scientific importance: Is the manuscript an original and important contribution to its field?

Acceptable

General interest: Is the paper of sufficient general interest?

Good

Quality of the paper: Is the overall quality of the paper suitable?

Marginal

Is the length of the paper justified?

Yes

Should the paper be seen by a specialist statistical reviewer?

Yes

Do you have any concerns about statistical analyses in this paper? If so, please specify them explicitly in your report.

Yes

It is a condition of publication that authors make their supporting data, code and materials available - either as supplementary material or hosted in an external repository. Please rate, if applicable, the supporting data on the following criteria.

Is it accessible?

No

Is it clear?

Yes

Is it adequate?

Yes

Do you have any ethical concerns with this paper?

No

Comments to the Author

Clearly written and explained although the conclusions contain a substantial amount of claims which are not justified (see below).

54-55 This brief summary of % plant extinctions across 3 mass extinctions is fundamentally misleading. The high rates of extinction from the TJ are based on McElwain's figures from NE Greenland - and is just a figure for local extirpation. Elsewhere in the world the extinction rate was much lower. Similar problem for the GLB data. Meanwhile, the [5] reference is based on a global database compilation which failed to do much taxonomic vetting and is deeply flawed.

114-119: The data is grouped by stratigraphic formation - so is there no time transgressive aspect to these units? Formations are not usually used as time bins. Also, the time duration of these units varies enormously - the Upper Guodikeng likely lasted <1 million years whilst the Huangshangji would have lasted more than 10 times as long. It would be interesting to see diversity versus duration of time bin plots, even if only in supplementary materials.

Geological setting: there needs to be some discussion of climate changes [if any] in the Junggar Basin over this long span of time.

122-30: this paragraph is the crux of the paper, and it is beyond my pay grade to say if the exponential - power law modelling is valid or not, I presume it is, but I'd have liked to know how the analysis incorporates missing elements of the communities (e.g. insects will not be represented in anything like their true importance). Also, how is abundance/occurrence accounted for, if at all? Some species may be extremely rare and so any links in the food webs will be trivial.

159: "tropic"!

Fundamentally the extinction model assumes diversity is controlled by primary productivity because the extinctions are modeled by reducing productivity. Diversity does not equal productivity in the oceans, but is more clearly linked on land I guess? And how is primary productivity assessed, not by plant diversity I presume, is it just assumed?

211-218: both freshwater and terrestrial communities are modelled together? They're clearly different (look at the fundamentally different response during the KPg event) - hence they plot in

different fields in figure 3, it would have been nice if they had not been lumped together.

The description here talks about the two trophic space areas and yet for nearly half the time plots in figure 3 only one space can be seen – why?

217: “no trophic space innovation was lost” – where can we see this in the plots?

229: “extinction affected communities”: why is the Capitanian considered extinction affected – the extinction happened in the latter part of this interval and so surely the Wuchiapingian community was the affected one? I also wonder what community they are assessing with their Capitanian dataset because it will include pre and post-extinction species.

Here we are also told the Rhaetian community was extinction affected although later (line 272) we are told this interval was “the prelude to the T-J extinction”?? – so why does the collapse threshold change before the terminal mass extinction? I suspect it is because of the enormously different durations of the Norian and Rhaetian – the former is ~5 times longer than the latter. Hence the diversity in the former could well be five times higher. However, later, even the existence of the TJ mass extinction is questioned (lines 370-1). Ideas about this mass extinction seem to evolve through the manuscript – they’re certainly not consistent.

298: “Here, we discovered...” None of this was actually discovered, it was already well known before – this is a meta-analysis of previously published work after all.

303: community size? An odd phrase, even if there were only a few species they could be present in huge numbers. “Community diversity” is better.

310: “revealed major disruptions” – the modeling did not reveal this because it was assumed in the modeling. Getting a handle on terrestrial biomass/productivity is not possible – the quoted references equate diversity with biomass but do not assess productivity in any way. Modeling like this can suggest/guess that productivity has collapsed – it’s a reasonable thing to model - but no more.

388: “relative magnitudes” – of what? Lava volumes?

390-4: this is all a bit vague and waffly. There’s talk of extinctions with sufficient magnitude to remove guilds – which one was this? The PT extinction has just been noted to have not removed any guilds in freshwater settings?

398-9: “The G-L and T-J extinctions were each preceded by low-stability communities” but it has just been noted that the G-L extinction actually happened within the Capitanian – so the “low stability” community will likely be the post-extinction community intermixed with the pre-extinction one. The so-called T-J pre-extinction community is likely just an artifact of different time bins (see above).

Review form: Reviewer 2

Recommendation

Major revision is needed (please make suggestions in comments)

Scientific importance: Is the manuscript an original and important contribution to its field?

Good

General interest: Is the paper of sufficient general interest?

Good

Quality of the paper: Is the overall quality of the paper suitable?

Acceptable

Is the length of the paper justified?

Yes

Should the paper be seen by a specialist statistical reviewer?

No

Do you have any concerns about statistical analyses in this paper? If so, please specify them explicitly in your report.

Yes

It is a condition of publication that authors make their supporting data, code and materials available - either as supplementary material or hosted in an external repository. Please rate, if applicable, the supporting data on the following criteria.

Is it accessible?

Yes

Is it clear?

Yes

Is it adequate?

Yes

Do you have any ethical concerns with this paper?

No

Comments to the Author

General comments: In this manuscript, the authors use a modeling approach (Cascading Extinctions on Graphs; CEG) to examine changes in community stability over time in the context of three major mass extinction events (the end-Guadalupian, the Permo-Triassic, and the Triassic-Jurassic). In some ways this paper is very exciting. The CEG model has been used extensively to examine Permo-Triassic tetrapod communities in southern Africa, and a now well-established narrative of the changes those communities underwent (in the context of the model) has been built up. This paper is the first application of the approach to terrestrial Permo-Triassic communities in a very different geographic area, and it is very interesting to see some of the same patterns of change emerging. The paper also applies the model to a set of communities spanning a longer time interval and more mass extinctions than has been the case in southern Africa, and this helps to contextualize some of the results seen in both places. In particular, this longer time interval emphasizes the uniqueness to the Permo-Triassic extinction compared to the others. These are important results. However, the manuscript also has some issues that I think need to be addressed before it is ready for publication.

1) At a broad level, I think there is a big missed opportunity in this manuscript. The small hint in the title notwithstanding, in reading this manuscript you would think that the communities here are basically the same as those studied in southern Africa, and I bet most readers would come away thinking that the big message is that tetrapod communities in Africa and China were affected by the PTME in the same way. However, the communities here actually differ considerably from the African ones in the sense that they have a much more heavily-sampled aquatic invertebrate fauna. That's potentially a very significant difference, but it hardly gets mentioned anywhere. The fact that this different kind of community (or at least differently sampled community) shows many of the same kinds of changes through the Permian and Triassic strikes me as an important point because it implies that different types of communities

were effected in much the same way by the PTME or that very different sampling regimes applied to more or less the same underlying terrestrial community will recover a consistent signal of ecosystem change. In my mind, those results are at least as important as the observation that overall patterns of stability over time are similar here to what's observed in southern Africa, and they really deserve some consideration in the manuscript. Indeed, they would constitute a shift from "more of the same" to "unique insights made possible by this new dataset".

2) Something I bring up in many of my comments below is that I'm surprised that the authors don't apply the approach of Roopnarine et al. (2018) for quantifying differences in the CEG results (i.e., using a combination of intercept and Hill's slope). The approach here considers a rough equivalent of only one of those parameters, but there's relevant variation in the results that would and should be captured by the other. Indeed, the Roopnarine et al. (2018) paper is probably the most relevant to this manuscript of all the CEG papers to date, both in terms of methods and conclusions, but it gets very little attention. Beyond including a method that would strengthen many of the results, I think the general message of that paper would further bolster many of the conclusions of this one.

Beyond these issues, there's some other methodological items that I think need to be addressed (e.g., discussion of how insect richness was estimated; some additions to the rarefaction analyses), areas where some clarifications are needed, changes to figures, and a few places where I think the authors misrepresent some of their results and those of other relevant papers (not in a malicious or intentionally misleading way). These all strike me as more minor problems in terms of scope, but they definitely should be addressed.

Line 26 to 30: This sentence seems needlessly complex. I recommend dividing it up into at least two.

Line 32: the lower in lower Permian should not be capitalized. Also, if word count allows, I recommend adding "the" before both lowers. Finally, change to shows (its subject, modelling, is singular).

Lines 46-47: remove any and insert mechanisms after killing

Line 56: I guess this sort of depends on what you consider similar. Of the three, continental configuration might be the most similar, but there were some fairly substantial changes in things like atmospheric CO₂ and O₂ levels, and global temperatures over this interval.

Lines 55-56: it would be good to have some references for the differing levels of marine vs. terrestrial diversity at the different times. Also, this is kind of a run-on sentence. I recommend splitting it up into two where the colon currently is.

Line 74: "trophic food web" seems a little redundant

Line 77: change to graphs

Line 90: change from to during

Line 92: examples of biostratigraphic and geochronology refs would be good to add here

Line 102: change were to are. Also, a more general comment about the methods. Only a couple of your communities have insects actually preserved in them (Table S1), but I see you have insects as part of all of your food webs (Table S2). The richnesses of the guilds vary across a fairly narrow range, so I assume you are estimating insect richness somehow (perhaps following the method used in previous CEG work in on African and North American paleocommunities). You should state that you are doing this, and note how you are making the estimates.

Line 110: I recommend using metanetwork instead of matrix for consistency with previous work using the CEG framework

Line 111: In previous works applying CEG to Permo-Triassic tetrapod communities, some basic rule were applied to structure interactions between guilds. Based on Fig. 1, it looks like those rules, or a slightly modified version, were applied here as well. I think it would be good to note that somewhere, either here in the methods or in the supplement.

line 112: change to early Permian

Line 130: If possible, include a couple citations to help support the observation that species level network variation doesn't strongly affect the results in most cases.

Lines 133-138: Is this actually in the right place? It seems to be presenting results, and presenting them before you actually describe the methods being used.

Line 149: change to prey. Also here, and in line 152, I recommend using metanetwork for the guild-level food web, similar to my earlier comment.

Line 160: you seem to be missing a parenthesis

Line 160-162: I kind of disagree with this statement. CEG doesn't directly test for variation in structural composition or guild richnesses. Those are input data for the model inasmuch as communities being examined in a set of analyses vary in those properties. Instead, it models the effects that those differences have on the response to the applied perturbations. The NMDS analyses you describe above, and the network properties to measure provide a further quantitative description of these differences, but again don't really represent a test in the sense of trying to falsify a hypothesis. I think a better way to describe things is to say that your combined approach (i.e., NMDS, network metrics, CEG results) allow you to describe how the structures of the communities differ, and then determine how those differences translate into community performance in the face of perturbations. The latter part of this could be construed as a test of whether the described differences have a significant effect on community performance.

Line 165: change was to were

Line 180: One of the long-running issues of work in the CEG framework has been how to best quantify aspects of the model output for comparison across communities. Change point analysis is one way to do this, but I also thought that the combination of looking at intercept and Hill's Slope used in Roopnarine et al. 2018 was effective. Indeed, it captures an aspect of performance (intercept) that is not considered in the current analysis, but that shows considerable variation in the southern African Permo-Triassic communities in the Roopnarine et al. paper. Is there a reason why that approach wasn't used here? If possible, I would suggest using it at least as a complement to the change point analyses.

Fig. 1: I recommend putting quotes around *Chasmatosuchus*; that seems to be current practice in the archosauromorph literature. Also I think the photo of the specimen will be a bit harder for non-specialists to read than the others, since the face isn't as obvious.

Fig. 2: The color scheme used definitely needs to be changed to make sure it is accessible to readers with colorblindness.

Line 213: I think you mean to call out fig S2 here.

Line 217: I recommend noting, maybe parenthetically, that such innovation would be manifested on the plots in Fig. 3 as the rise of a new peak on the landscape.

Fig. 3: Here again, I think you should avoid using a a color ramp that uses red-green differences so strongly.

Line 230: change to periods; change to late Permian

Line 232-233: I think it would be better to say that diversity is correlated with connectance (i.e., omit the “changes”)

Line 243: This variation in response to low perturbations isn't really captured in Fig. 5, but in some ways it's one of the most interesting parallels to previous results from southern Africa. This again makes me think using the intercept/Hill's slope approach used by Roopnarine et al. (2018), with a bivariate plot showing communities in that space, would provide a fuller description of the differences among communities.

Line 289: change did to does

line 294: change functionally to functional

Line 310-313: I disagree with these statements. The approach you're using here cannot reveal anything directly about changes in primary productivity because the units of productivity used in the model are estimates based on herbivore diversity (i.e., they are not actual observations about the floras at the different times). At best they can tell you that the consumer communities at a given time might have required more or less primary productivity to be available than at some other time, but that is a very different statement. It only sets a minimum floor on required productivity, but it doesn't say anything about what the actual available productivity was. Likewise, it was never claimed in any of the papers on southern African Permo-Triassic communities that actual productivity levels declined. A more accurate thing to say here would be that the observed changes in animal diversity suggest a decline in the amount of productivity that is needed to support the communities, and that in turn is perhaps consistent with the fact that paleobotanical evidence suggests that suggests a drop in plant diversity at this time. You can also note that a similar implied drop in productivity requires was observed in the African communities.

Line 317: This is kind of a weak statement. I think you could strengthen it by talking a bit more about the potential implications of the diversity decline. I understand that to be a change from a diverse, more forested environment to a more open habitat dominated by a few species of plants such as *Pleuromaia*. Describing that transition in a little more detail would help to make the expectation of reduced productivity more obvious.

Line 322: I think some readers would find the inability to localize the cause of the change to one or more of the variables unsatisfying because they are unfamiliar with the non-linear dynamics of the CEG model, which in turn make such predictions quite difficult. I think noting this difficulty will help to reduce that concern and explain why there isn't an obvious factor you can point to as the cause.

Line 330: I recommend moving the citation for ref 38 to the end of this sentence.

Line 337: I know it's a drum I'm beating a lot in this review, but considering intercept here would be good as well. It too has returned to “normal” in the CEG plots by the Middle Triassic.

Line 339: I would expand this to Southern Africa more generally, following the results of Roopnarine et al. (2018), and recommend citing that paper here. In many ways, it is methodologically the most directly comparable of all the CEG work to your manuscript, and it helps to further establish the apparently broad geographic pattern of recovery in the Middle Triassic.

Line 340-345: I think there's a couple ideas here that should be unpacked a little more for clarity. 1) High connectivity, or the emergence of a high connectivity guild, might help to stabilize the community in the model because of changes in network dynamics (and perhaps regardless of whether the guild was itself a generalist or a specialist that happened to be preyed on by many other guilds); 2) generalist trophic habits for the guild that would tend to make it less likely to be fatally impacted when the community is disturbed. It seems like you're implying that the notostracans were generalists (and thus had lots of food sources they could draw upon when the community was disturbed) and that they were preyed upon by a lot of other guilds (and thus were a reliable food source for other taxa), so a bit of both 1 and 2. Do you think that same combination applied to the other highly connected guilds you mention?

Line 377: That's not exactly how I would characterize the message of the Mitchell et al. paper. Instead, their result was more that the Maastrichtian communities in particular were less resistant on average than the Campanian ones. It is that lower resistance of the Maastrichtian communities that has implications for the K-Pg extinction (i.e., it might not have been quite as bad if the asteroid hit in the Campanian...).

Line 397: Change to: "Our results indicate three significant decreases in paleocommunity stability in Xinjiang, coinciding with the G-L, P-Tr, and T-J extinctions."

Line 388: I'm not sure how much space is available to do it, but I think this is an idea that should be developed further. The pattern of decreasing stability leading up to an extinction vs. following an extinction seems significant, and I have a hard time thinking that just the size of the event is the only thing at play (as opposed to the rate of environmental changes, details of environmental effects, etc.). You might be right that it's just an issue of magnitude, but you should present more evidence to help support this claim.

Line 403: insert that after confirmed

Line 438: change to Botha-Brink

Supplement, Sampling issues paragraph 1: I think you mean to call out figure S3 here.

Supplement, Sampling issues paragraph 3, line 2: change to communities.

Supplement, Sampling issues general comment: This approach refutes the idea that sampling is an issue if the sampling probability is more or less the same across all guilds and across all time intervals. However, looking at table S1, there are some patterns that make me think this is definitely not the case (e.g., insects only being preserved in the Late Triassic, the near complete absence of bivalves in the Early Triassic, despite their considerable diversity at other times). Therefore, in addition to your random re-sampling, it might be good to do some more targeted manipulations to see what effect non-random patterns of sampling might have (e.g., what are your Permian communities like if there are no or very few bivalves in them?). These sorts of experiments might also be helpful in nailing down some more specific effects of particular guilds, such as the highly connected ones you mention in the main text.

Review form: Reviewer 3

Recommendation

Accept as is

Scientific importance: Is the manuscript an original and important contribution to its field?

Excellent

General interest: Is the paper of sufficient general interest?

Excellent

Quality of the paper: Is the overall quality of the paper suitable?

Excellent

Is the length of the paper justified?

Yes

Should the paper be seen by a specialist statistical reviewer?

No

Do you have any concerns about statistical analyses in this paper? If so, please specify them explicitly in your report.

No

It is a condition of publication that authors make their supporting data, code and materials available - either as supplementary material or hosted in an external repository. Please rate, if applicable, the supporting data on the following criteria.

Is it accessible?

Yes

Is it clear?

Yes

Is it adequate?

Yes

Do you have any ethical concerns with this paper?

No

Comments to the Author

This study analyzes the changes in terrestrial and freshwater communities across three mass extinctions, namely the end-Guadalupian, end-Permian and end-Triassic mass extinctions. The authors use data from northern Xinjiang, North West China to assess changes in species diversity, guilds and trophic space. This is a comprehensive, well-documented paper with globally relevant results. The most significant result was confirming the uniqueness and extent of the end-Permian mass extinction compared to the other two crises, although they also found some generalities in all three events, which is also interesting. This is a well-written, well-explained documentation of new data that makes for a significant contribution towards mass extinction studies, and thus, I recommend publication as is. I picked up a few very minor typos in the text, which I attach as a separate document.

Decision letter (RSPB-2020-2404.R0)

26-Oct-2020

Dear Professor Chen:

I am writing to inform you that your manuscript RSPB-2020-2404 entitled "Ecological dynamics of terrestrial and freshwater ecosystems across three mid-Phanerozoic mass extinctions from northwest China" has, in its current form, been rejected for publication in Proceedings B.

This action has been taken on the advice of referees, who have recommended that substantial revisions are necessary. With this in mind we would be happy to consider a resubmission, provided the comments of the referees are fully addressed. However please note that this is not a provisional acceptance.

Sincerely,
Dr John Hutchinson, Editor
mailto:proceedingsb@royalsociety.org

Associate Editor
Comments to Author:
Dear Authors

You will see that all reviewers identify a lot of strengths in this contribution and find it highly valuable. However, there is a number of points that need clarification or justification and perhaps some points that is missed in terms of the broader comparison between China and South Africa. Please consider these suggestions and the manuscript will be considered for a second round of review.

Reviewer(s)' Comments to Author:
Referee: 1
Comments to the Author(s)

Clearly written and explained although the conclusions contain a substantial amount of claims which are not justified (see below).

54-55 This brief summary of % plant extinctions across 3 mass extinctions is fundamentally misleading. The high rates of extinction from the TJ are based on McElwain's figures from NE Greenland - and is just a figure for local extirpation. Elsewhere in the world the extinction rate was much lower. Similar problem for the GLB data. Meanwhile, the [5] reference is based on a global database compilation which failed to do much taxonomic vetting and is deeply flawed.

114-119: The data is grouped by stratigraphic formation – so is there no time transgressive aspect to these units? Formations are not usually used as time bins. Also, the time duration of these units varies enormously – the Upper Guodikeng likely lasted <1 million years whilst the Huangshangji would have lasted more than 10 times as long. It would be interesting to see diversity versus duration of time bin plots, even if only in supplementary materials.

Geological setting: there needs to be some discussion of climate changes [if any] in the Junggar Basin over this long span of time.

122-30: this paragraph is the crux of the paper, and it is beyond my pay grade to say if the exponential - power law modelling is valid or not, I presume it is, but I'd have liked to know how the analysis incorporates missing elements of the communities (e.g. insects will not be represented in anything like their true importance). Also, how is abundance/occurrence accounted for, if at all? Some species may be extremely rare and so any links in the food webs will be trivial.

159: “tropic”!

Fundamentally the extinction model assumes diversity is controlled by primary productivity because the extinctions are modeled by reducing productivity. Diversity does not equal productivity in the oceans, but is more clearly linked on land I guess? And how is primary productivity assessed, not by plant diversity I presume, is it just assumed?

211-218: both freshwater and terrestrial communities are modelled together? They're clearly different (look at the fundamentally different response during the KPg event) – hence they plot in different fields in figure 3, it would have been nice if they had not been lumped together. The description here talks about the two trophic space areas and yet for nearly half the time plots in figure 3 only one space can be seen – why?

217: “no trophic space innovation was lost” – where can we see this in the plots?

229: “extinction affected communities”: why is the Capitanian considered extinction affected – the extinction happened in the latter part of this interval and so surely the Wuchiapingian community was the affected one? I also wonder what community they are assessing with their Capitanian dataset because it will include pre and post-extinction species.

Here we are also told the Rhaetian community was extinction affected although later (line 272) we are told this interval was “the prelude to the T-J extinction”?? – so why does the collapse threshold change before the terminal mass extinction? I suspect it is because of the enormously different durations of the Norian and Rhaetian – the former is ~5 times longer than the latter. Hence the diversity in the former could well be five times higher. However, later, even the existence of the TJ mass extinction is questioned (lines 370-1). Ideas about this mass extinction seem to evolve through the manuscript – they're certainly not consistent.

298: “Here, we discovered....” None of this was actually discovered, it was already well known before – this is a meta-analysis of previously published work after all.

303: community size? An odd phrase, even if there were only a few species they could be present in huge numbers. “Community diversity” is better.

310: “revealed major disruptions” – the modeling did not reveal this because it was assumed in the modeling. Getting a handle on terrestrial biomass/productivity is not possible – the quoted references equate diversity with biomass but do not assess productivity in any way. Modeling like this can suggest/guess that productivity has collapsed – it's a reasonable thing to model - but no more.

388: “relative magnitudes” – of what? Lava volumes?

390-4: this is all a bit vague and waffly. There's talk of extinctions with sufficient magnitude to remove guilds – which one was this? The PT extinction has just been noted to have not removed any guilds in freshwater settings?

398-9: “The G-L and T-J extinctions were each preceded by low-stability communities” but it has just been noted that the G-L extinction actually happened within the Capitanian – so the “low stability” community will likely be the post-extinction community intermixed with the pre-extinction one. The so-called T-J pre-extinction community is likely just an artifact of different time bins (see above).

Referee: 2

Comments to the Author(s)

General comments: In this manuscript, the authors use a modeling approach (Cascading Extinctions on Graphs; CEG) to examine changes in community stability over time in the context of three major mass extinction events (the end-Guadalupian, the Permo-Triassic, and the Triassic-Jurassic). In some ways this paper is very exciting. The CEG model has been used extensively to examine Permo-Triassic tetrapod communities in southern Africa, and a now well-established narrative of the changes those communities underwent (in the context of the model) has been built up. This paper is the first application of the approach to terrestrial Permo-Triassic communities in a very different geographic area, and it is very interesting to see some of the same patterns of change emerging. The paper also applies the model to a set of communities spanning a longer time interval and more mass extinctions than has been the case in southern Africa, and this helps to contextualize some of the results seen in both places. In particular, this longer time interval emphasizes the uniqueness to the Permo-Triassic extinction compared to the others. These are important results. However, the manuscript also has some issues that I think need to be addressed before it is ready for publication.

1) At a broad level, I think there is a big missed opportunity in this manuscript. The small hint in the title notwithstanding, in reading this manuscript you would think that the communities here are basically the same as those studied in southern Africa, and I bet most readers would come away thinking that the big message is that tetrapod communities in Africa and China were affected by the PTME in the same way. However, the communities here actually differ considerably from the African ones in the sense that they have a much more heavily-sampled aquatic invertebrate fauna. That's potentially a very significant difference, but it hardly gets mentioned anywhere. The fact that this different kind of community (or at least differently sampled community) shows many of the same kinds of changes through the Permian and Triassic strikes me as an important point because it implies that different types of communities were effected in much the same way by the PTME or that very different sampling regimes applied to more or less the same underlying terrestrial community will recover a consistent signal of ecosystem change. In my mind, those results are at least as important as the observation that overall patterns of stability over time are similar here to what's observed in southern Africa, and they really deserve some consideration in the manuscript. Indeed, they would constitute a shift from “more of the same” to “unique insights made possible by this new dataset”.

2) Something I bring up in many of my comments below is that I'm surprised that the authors don't apply the approach of Roopnarine et al. (2018) for quantifying differences in the CEG results (i.e., using a combination of intercept and Hill's slope). The approach here considers a rough equivalent of only one of those parameters, but there's relevant variation in the results that would and should be captured by the other. Indeed, the Roopnarine et al. (2018) paper is probably the most relevant to this manuscript of all the CEG papers to date, both in terms of methods and conclusions, but it gets very little attention. Beyond including a method that would strengthen many of the results, I think the general message of that paper would further bolster many of the conclusions of this one.

Beyond these issues, there's some other methodological items that I think need to be addressed (e.g., discussion of how insect richness was estimated; some additions to the rarefaction analyses), areas where some clarifications are needed, changes to figures, and a few places where I think the authors misrepresent some of their results and those of other relevant papers (not in a malicious or intentionally misleading way). These all strike me as more minor problems in terms of scope, but they definitely should be addressed.

Line 26 to 30: This sentence seems needlessly complex. I recommend dividing it up into at least two.

Line 32: the lower in lower Permian should not be capitalized. Also, if word count allows, I recommend adding "the" before both lowers. Finally, change to shows (its subject, modelling, is singular).

Lines 46-47: remove any and insert mechanisms after killing

Line 56: I guess this sort of depends on what you consider similar. Of the three, continental configuration might be the most similar, but there were some fairly substantial changes in things like atmospheric CO₂ and O₂ levels, and global temperatures over this interval.

Lines 55-56: it would be good to have some references for the differing levels of marine vs. terrestrial diversity at the different times. Also, this is kind of a run-on sentence. I recommend splitting it up into two where the colon currently is.

Line 74: "trophic food web" seems a little redundant

Line 77: change to graphs

Line 90: change from to during

Line 92: examples of biostratigraphic and geochronology refs would be good to add here

Line 102: change were to are. Also, a more general comment about the methods. Only a couple of your communities have insects actually preserved in them (Table S1), but I see you have insects as part of all of your food webs (Table S2). The richnesses of the guilds vary across a fairly narrow range, so I assume you are estimating insect richness somehow (perhaps following the method used in previous CEG work in on African and North American paleocommunities). You should state that you are doing this, and note how you are making the estimates.

Line 110: I recommend using metanetwork instead of matrix for consistency with previous work using the CEG framework

Line 111: In previous works applying CEG to Permo-Triassic tetrapod communities, some basic rule were applied to structure interactions between guilds. Based on Fig. 1, it looks like those rules, or a slightly modified version, were applied here as well. I think it would be good to note that somewhere, either here in the methods or in the supplement.

line 112: change to early Permian

Line 130: If possible, include a couple citations to help support the observation that species level network variation doesn't strongly affect the results in most cases.

Lines 133-138: Is this actually in the right place? It seems to be presenting results, and presenting them before you actually describe the methods being used.

Line 149: change to prey. Also here, and in line 152, I recommend using metanetwork for the guild-level food web, similar to my earlier comment.

Line 160: you seem to be missing a parenthesis

Line 160-162: I kind of disagree with this statement. CEG doesn't directly test for variation in structural composition or guild richnesses. Those are input data for the model inasmuch as communities being examined in a set of analyses vary in those properties. Instead, it models the effects that those differences have on the response to the applied perturbations. The NMDS analyses you describe above, and the network properties to measure provide a further quantitative description of these differences, but again don't really represent a test in the sense of trying to falsify a hypothesis. I think a better way to describe things is to say that your combined approach (i.e., NMDS, network metrics, CEG results) allow you to describe how the structures of the communities differ, and then determine how those differences translate into community performance in the face of perturbations. The latter part of this could be construed as a test of whether the described differences have a significant effect on community performance.

Line 165: change was to were

Line 180: One of the long-running issues of work in the CEG framework has been how to best quantify aspects of the model output for comparison across communities. Change-point analysis is one way to do this, but I also thought that the combination of looking at intercept and Hill's Slope used in Roopnarine et al. 2018 was effective. Indeed, it captures an aspect of performance (intercept) that is not considered in the current analysis, but that shows considerable variation in the southern African Permo-Triassic communities in the Roopnarine et al. paper. Is there a reason why that approach wasn't used here? If possible, I would suggest using it at least as a complement to the change-point analyses.

Fig. 1: I recommend putting quotes around *Chasmatosuchus*; that seems to be current practice in the archosauromorph literature. Also I think the photo of the specimen will be a bit harder for non-specialists to read than the others, since the face isn't as obvious.

Fig. 2: The color scheme used definitely needs to be changed to make sure it is accessible to readers with colorblindness.

Line 213: I think you mean to call out fig S2 here.

Line 217: I recommend noting, maybe parenthetically, that such innovation would be manifested on the plots in Fig. 3 as the rise of a new peak on the landscape.

Fig. 3: Here again, I think you should avoid using a color ramp that uses red-green differences so strongly.

Line 230: change to periods; change to late Permian

Line 232-233: I think it would be better to say that diversity is correlated with connectance (i.e., omit the "changes")

Line 243: This variation in response to low perturbations isn't really captured in Fig. 5, but in some ways it's one of the most interesting parallels to previous results from southern Africa. This again makes me think using the intercept/Hill's slope approach used by Roopnarine et al. (2018), with a bivariate plot showing communities in that space, would provide a fuller description of the differences among communities.

Line 289: change did to does

line 294: change functionally to functional

Line 310-313: I disagree with these statements. The approach you're using here cannot reveal anything directly about changes in primary productivity because the units of productivity used in the model are estimates based on herbivore diversity (i.e., they are not actual observations about the floras at the different times). At best they can tell you that the consumer communities at a given time might have required more or less primary productivity to be available than at some other time, but that is a very different statement. It only sets a minimum floor on required productivity, but it doesn't say anything about what the actual available productivity was. Likewise, it was never claimed in any of the papers on southern African Permo-Triassic communities that actual productivity levels declined. A more accurate thing to say here would be that the observed changes in animal diversity suggest a decline in the amount of productivity that is needed to support the communities, and that in turn is perhaps consistent with the fact that paleobotanical evidence suggests that suggests a drop in plant diversity at this time. You can also note that a similar implied drop in productivity requires was observed in the African communities.

Line 317: This is kind of a weak statement. I think you could strengthen it by talking a bit more about the potential implications of the diversity decline. I understand that to be a change from a diverse, more forested environment to a more open habitat dominated by a few species of plants such as *Pleuromaia*. Describing that transition in a little more detail would help to make the expectation of reduced productivity more obvious.

Line 322: I think some readers would find the inability to localize the cause of the change to one or more of the variables unsatisfying because they are unfamiliar with the non-linear dynamics of the CEG model, which in turn make such predictions quite difficult. I think noting this difficulty will help to reduce that concern and explain why there isn't an obvious factor you can point to as the cause.

Line 330: I recommend moving the citation for ref 38 to the end of this sentence.

Line 337: I know it's a drum I'm beating a lot in this review, but considering intercept here would be good as well. It too has returned to "normal" in the CEG plots by the Middle Triassic.

Line 339: I would expand this to Southern Africa more generally, following the results of Roopnarine et al. (2018), and recommend citing that paper here. In many ways, it is methodologically the most directly comparable of all the CEG work to your manuscript, and it helps to further establish the apparently broad geographic pattern of recovery in the Middle Triassic.

Line 340-345: I think there's a couple ideas here that should be unpacked a little more for clarity. 1) High connectivity, or the emergence of a high connectivity guild, might help to stabilize the community in the model because of changes in network dynamics (and perhaps regardless of whether the guild was itself a generalist or a specialist that happened to be preyed on by many other guilds); 2) generalist trophic habits for the guild that would tend to make it less likely to be fatally impacted when the community is disturbed. It seems like your implying that the notostracans were generalists (and thus had lots of food sources they could draw upon when the community was disturbed) and that they were preyed upon by a lot of other guilds (and thus were a reliable food source for other taxa), so a bit of both 1 and 2. Do you think that same combination applied to the other highly connected guilds you mention?

Line 377: That's not exactly how I would characterize the message of the Mitchell et al. paper. Instead, their result was more that the Maastrichtian communities in particular were less resistant on average than the Campanian ones. It is that lower resistance of the Maastrichtian communities that has implications for the K-Pg extinction (i.e., it might not have been quite as bad if the asteroid hit in the Campanian...).

Line 397: Change to: “Our results indicate three significant decreases in paleocommunity stability in Xinjiang, coinciding with the G-L, P-Tr, and T-J extinctions.”

Line 388: I’m not sure how much space is available to do it, but I think this is an idea that should be developed further. The pattern of decreasing stability leading up to an extinction vs. following an extinction seems significant, and I have a hard time thinking that just the size of the event is the only thing at play (as opposed to the rate of environmental changes, details of environmental effects, etc.). You might be right that it’s just an issue of magnitude, but you should present more evidence to help support this claim.

Line 403: insert that after confirmed

Line 438: change to Botha-Brink

Supplement, Sampling issues paragraph 1: I think you mean to call out figure S3 here.

Supplement, Sampling issues paragraph 3, line 2: change to communities.

Supplement, Sampling issues general comment: This approach refutes the idea that sampling is an issue if the sampling probability is more or less the same across all guilds and across all time intervals. However, looking at table S1, there are some patterns that make me think this is definitely not the case (e.g., insects only being preserved in the Late Triassic, the near complete absence of bivalves in the Early Triassic, despite their considerable diversity at other times). Therefore, in addition to your random re-sampling, it might be good to do some more targeted manipulations to see what effect non-random patterns of sampling might have (e.g., what are your Permian communities like if there are no or very few bivalves in them?). These sorts of experiments might also be helpful in nailing down some more specific effects of particular guilds, such as the highly connected ones you mention in the main text.

Referee: 3

Comments to the Author(s)

This study analyzes the changes in terrestrial and freshwater communities across three mass extinctions, namely the end-Guadalupian, end-Permian and end-Triassic mass extinctions. The authors use data from northern Xinjiang, North West China to assess changes in species diversity, guilds and trophic space. This is a comprehensive, well-documented paper with globally relevant results. The most significant result was confirming the uniqueness and extent of the end-Permian mass extinction compared to the other two crises, although they also found some generalities in all three events, which is also interesting. This is a well-written, well-explained documentation of new data that makes for a significant contribution towards mass extinction studies, and thus, I recommend publication as is. I picked up a few very minor typos in the text, which I attach as a separate document.

Author's Response to Decision Letter for (RSPB-2020-2404.R0)

See Appendix A.

RSPB-2021-0148.R0

Review form: Reviewer 2

Recommendation

Accept with minor revision (please list in comments)

Scientific importance: Is the manuscript an original and important contribution to its field?

Excellent

General interest: Is the paper of sufficient general interest?

Good

Quality of the paper: Is the overall quality of the paper suitable?

Excellent

Is the length of the paper justified?

Yes

Should the paper be seen by a specialist statistical reviewer?

No

Do you have any concerns about statistical analyses in this paper? If so, please specify them explicitly in your report.

No

It is a condition of publication that authors make their supporting data, code and materials available - either as supplementary material or hosted in an external repository. Please rate, if applicable, the supporting data on the following criteria.

Is it accessible?

Yes

Is it clear?

Yes

Is it adequate?

Yes

Do you have any ethical concerns with this paper?

No

Comments to the Author

General comments: This is the second time I've reviewed this manuscript, and I think the authors have done a good job overall in responding to my suggestions and those of the other reviewers. I have a few suggested edits below, but they're mostly pretty minor that should be easy to fix. My only remaining sticking point deals with the comment in line 349 about the modeling results suggesting a disruption of primary productivity. I still don't think that's the case because that type of disruption is assumed by the modeled scenario in the first place (i.e., the model results are generated by removing producers). The authors note several other lines of evidence to suggest that such a disruption of producers occurred in Xinjiang near the PTB, which is good, but those are the data sources that need to be cited for that inference. All the CEG model can tell you is what the potential effects of such disturbances to the community were, not whether the disturbance itself happened.

Line 31: change to: and is flanked

line 80: I think 'here' can be removed

line 112: early (and middle and late) shouldn't be capitalized for the Permian (double-check throughout the manuscript that this is consistent)

line 112: delete the before northern

line 116: delete the before north. Also, I don't think north needs to be capitalized here.

Line 122: change were to are

line 136: change while to whereas

Line 145: I think this paragraph (lines 145–154) would actually fit better in the previous section (i.e., Database). Its current placement sort of interrupts the flow of information your providing about assembling the food webs.

line 146: I don't think you need to capitalize North

line 178: Would a better title for this section be something like "Modelling of Palaeocommunity Dynamics"?

Line 202: I'm not convinced that this paragraph (line 202–223) actually needs to be a distinct section. I think it would be better to incorporate it into the previous section, probably towards the middle of that section. That ways the logical progression would be 1) introduction to CEG (~ lines 179–194 in the current manuscript); 2) details of the application of the model and quantification of results (current section f); 3) comments about the combine approach (~lines 195–200 in the current version). There's a little bit of repetition between the two sections that would need to be dealt with, but that should be pretty simple to do.

Fig. 1: I think you misunderstood my previous comment here. You don't need quotes around all the taxon names in the photos, ust for "Chasmatosaurus" (e.g., there's no taxonomic issues for *Urumchia* that I am aware of that would necessitate it being in quotes).

Fig 4. Check the color scheme here for colorblindness accessibility. The red/green combination likely should be changed. Can you switch to the color palette used in Fig. 3 (which looks nice!).

Lines 337–340: This sentence seems kind of awkward. I think you should at least change the comma after amphibian to a semi-colon.

Line 349: I still disagree with this statement. The CEG models do not suggest (any more than they reveal) major disruptions in primary productivity. They only indicate that IF there were major disruptions in productivity, they likely would have resulted in increased sensitivities and decreased resistance. Whether such disruptions in productivity occurred (i.e., whether the scenario you are modeling is accurate) must be determined from other data, not the model. Similarly, the results for the Karoo do not imply a drop in productivity, they assume that scenario in the model. In the case of the Karoo, that scenario was chosen because at the time of writing, the prevailing idea for environmental change in the Karoo at the time included things like large plant die-offs, increased erosion, etc. This may seem like semantics, but to me there is a very fundamental distinction between what is model input vs. output. A way to deal with this might be to restructure this paragraph so that your comments about the paleobotanical record and the drop in animal diversity come first, and then present the (modeled) implications.

Line 384: I think you should note what the conclusions of those studies have been. That will provide more useful context for the statements you make in the rest of the paragraph.

Line 434: I'm not sure depleted is the right word here. Would "much lower threshold values" be better?

Review form: Reviewer 4 (Neil Brocklehurst)

Recommendation

Accept with minor revision (please list in comments)

Scientific importance: Is the manuscript an original and important contribution to its field?

Excellent

General interest: Is the paper of sufficient general interest?

Good

Quality of the paper: Is the overall quality of the paper suitable?

Good

Is the length of the paper justified?

Yes

Should the paper be seen by a specialist statistical reviewer?

No

Do you have any concerns about statistical analyses in this paper? If so, please specify them explicitly in your report.

No

It is a condition of publication that authors make their supporting data, code and materials available - either as supplementary material or hosted in an external repository. Please rate, if applicable, the supporting data on the following criteria.

Is it accessible?

Yes

Is it clear?

Yes

Is it adequate?

Yes

Do you have any ethical concerns with this paper?

No

Comments to the Author

This paper represents a thorough examination of ecosystem structure through three mass extinctions in a single area. The paper had already been through a round of reviews before it came to me. I read through the manuscript first but also had a quick look at the response to the previous reviews. On the whole I think this is an interesting paper which provides a new perspective not only on the specific ecosystems in question but also on mass extinctions in general. I think the authors have thoroughly responded to the reviewers' previous concerns and

have produced an excellent contribution. I have just a couple of method questions that would be good to clarify and then I then I think it would be ready to go.

Where you are using the method of Mitchell et al. to estimate insect richness, how do you then divide insects between predators, herbivores and omnivores? At random? Based on observed insect faunas?

Skull length: were species with no skull left out or was their size category estimated?

Perhaps in the supplement indicate which diet and size category you are assigning the vertebrate taxa?

Neil Brocklehurst

Decision letter (RSPB-2021-0148.R0)

09-Feb-2021

Dear Professor Chen:

Your manuscript has now been peer reviewed and the reviews have been assessed by an Associate Editor. The reviewers' comments (not including confidential comments to the Editor) and the comments from the Associate Editor are included at the end of this email for your reference. As you will see, the reviewers and the Editors have raised some concerns with your manuscript and we would like to invite you to revise your manuscript to address them.

Research ethics:

Use of animals and field studies:

It is a condition of publication that you make available the data and research materials supporting the results in the article (<https://royalsociety.org/journals/authors/author-guidelines/#data>). Datasets should be deposited in an appropriate publicly available repository and details of the associated accession number, link or DOI to the datasets must be included in the Data Accessibility section of the article (<https://royalsociety.org/journals/ethics-policies/data-sharing-mining/>). Reference(s) to datasets should also be included in the reference list of the article with DOIs (where available).

Ensure that "data" here includes code and all other files needed to replicate/re-use the study's raw data as per our open science/data policy.

Please submit a copy of your revised paper within three weeks. If we do not hear from you within this time your manuscript will be rejected. If you are unable to meet this deadline please let us know as soon as possible, as we may be able to grant a short extension.

Best wishes,
 Dr John Hutchinson, Editor
 mailto: proceedingsb@royalsociety.org

Associate Editor Board Member

Comments to Author:

Based on comments from the reviewers the manuscript is improved massively. There are still some minor issues that seems to make the findings and arguments inconsistent as highlighted by reviewer 2 in particular.

those should be addressed before a final consideration of the manuscript for publication

Reviewer(s)' Comments to Author:

Referee: 4

Comments to the Author(s).

This paper represents a thorough examination of ecosystem structure through three mass extinctions in a single area. The paper had already been through a round of reviews before it came to me. I read through the manuscript first but also had a quick look at the response to the previous reviews. On the whole I think this is an interesting paper which provides a new perspective not only on the specific ecosystems in question but also on mass extinctions in general. I think the authors have thoroughly responded to the reviewers' previous concerns and have produced an excellent contribution. I have just a couple of method questions that would be good to clarify and then I then I think it would be ready to go.

Where you are using the method of Mitchell et al. to estimate insect richness, how do you then divide insects between predators, herbivores and omnivores? At random? Based on observed insect faunas?

Skull length: were species with no skull left out or was their size category estimated?

Perhaps in the supplement indicate which diet and size category you are assigning the vertebrate taxa?

Neil Brocklehurst

Referee: 2

Comments to the Author(s).

General comments: This is the second time I've reviewed this manuscript, and I think the authors have done a good job overall in responding to my suggestions and those of the other reviewers. I have a few suggested edits below, but they're mostly pretty minor that should be easy to fix. My only remaining sticking point deals with the comment in line 349 about the modeling results suggesting a disruption of primary productivity. I still don't think that's the case because that type of disruption is assumed by the modeled scenario in the first place (i.e., the model results are generated by removing producers). The authors note several other lines of evidence to suggest that such a disruption of producers occurred in Xinjiang near the PTB, which is good, but those are the data sources that need to be cited for that inference. All the CEG model can tell you is what the potential effects of such disturbances to the community were, not whether the disturbance itself happened.

Line 31: change to: and is flanked

line 80: I think 'here' can be removed

line 112: early (and middle and late) shouldn't be capitalized for the Permian (double-check throughout the manuscript that this is consistent)

line 112: delete the before northern

line 116: delete the before north. Also, I don't think north needs to be capitalized here.

Line 122: change were to are

line 136: change while to whereas

Line 145: I think this paragraph (lines 145–154) would actually fit better in the previous section (i.e., Database). Its current placement sort of interrupts the flow of information your providing about assembling the food webs.

line 146: I don't think you need to capitalize North

line 178: Would a better title for this section be something like "Modelling of Palaeocommunity Dynamics"?

Line 202: I'm not convinced that this paragraph (line 202–223) actually needs to be a distinct section. I think it would be better to incorporate it into the previous section, probably towards the middle of that section. That ways the logical progression would be 1) introduction to CEG (~ lines 179–194 in the current manuscript); 2) details of the application of the model and quantification of results (current section f); 3) comments about the combine approach (~lines 195–200 in the current version). There's a little bit of repetition between the two sections that would need to be dealt with, but that should be pretty simple to do.

Fig. 1: I think you misunderstood my previous comment here. You don't need quotes around all the taxon names in the photos, ust for "Chasmatosaurus" (e.g., there's no taxonomic issues for *Urumchia* that I am aware of that would necessitate it being in quotes).

Fig 4. Check the color scheme here for colorblindness accessibility. The red/green combination likely should be changed. Can you switch to the color palette used in Fig. 3 (which looks nice!).

Lines 337–340: This sentence seems kind of awkward. I think you should at least change the comma after amphibian to a semi-colon.

Line 349: I still disagree with this statement. The CEG models do not suggest (any more than they reveal) major disruptions in primary productivity. They only indicate that IF there were major disruptions in productivity, they likely would have resulted in increased sensitivities and decreased resistance. Whether such disruptions in productivity occurred (i.e., whether the scenario you are modeling is accurate) must be determined from other data, not the model. Similarly, the results for the Karoo do not imply a drop in productivity, they assume that scenario in the model. In the case of the Karoo, that scenario was chosen because at the time of writing, the prevailing idea for environmental change in the Karoo at the time included things like large plant die-offs, increased erosion, etc. This may seem like semantics, but to me there is a very fundamental distinction between what is model input vs. output. A way to deal with this might be to restructure this paragraph so that your comments about the paleobotanical record and the drop in animal diversity come first, and then present the (modeled) implications.

Line 384: I think you should note what the conclusions of those studies have been. That will provide more useful context for the statements you make in the rest of the paragraph.

Line 434: I'm not sure depleted is the right word here. Would "much lower threshold values" be better?

Author's Response to Decision Letter for (RSPB-2021-0148.R0)

See Appendix B.

Decision letter (RSPB-2021-0148.R1)

22-Feb-2021

Dear Professor Chen

I am pleased to inform you that your manuscript entitled "Ecological dynamics of terrestrial and freshwater ecosystems across three mid-Phanerozoic mass extinctions from northwest China" has been accepted for publication in Proceedings B. Congratulations!!

Open Access

Paper charges

Sincerely,
Dr John Hutchinson

Editor, Proceedings B
mailto:proceedingsb@royalsociety.org

Associate Editor:

Comments to Author:

It seems that comments by referees have been addressed and that the manuscript is ready to move onwards to publication.

Reviewer(s)' Comments to Author:

Referee: 1

Comments to the Author(s)

Clearly written and explained although the conclusions contain a substantial amount of claims which are not justified (see below).

54-55 This brief summary of % plant extinctions across 3 mass extinctions is fundamentally misleading. The high rates of extinction from the TJ are based on McElwain's figures from NE Greenland – and is just a figure for local extirpation. Elsewhere in the world the extinction rate was much lower. Similar problem for the GLB data. Meanwhile, the [5] reference is based on a global database compilation which failed to do much taxonomic vetting and is deeply flawed.

Reply: Thanks. We delete these references, instead, terrestrial tetrapod genera extinction rate is applied to indicate these extinction events. The tetrapod extinction rates in the P-Tr and T-J extinctions follow the global data compiled by Benton et al. (2013, Palaeo-3). Global tetrapod extinction rate during the G-L transition still remains unclear. Here, the G-L extinction rates derived from the Karoo basin (Day et al., 2015, PRSB; Lucas, 2017, ESR) are applied to represent the G-L extinction. This is because the Karoo basin has the best Permian tetrapod records in the world.

114-119: The data is grouped by stratigraphic formation – so is there no time transgressive aspect to these units? Formations are not usually used as time bins. Also, the time duration of these units varies enormously – the Upper Guodikeng likely lasted <1 million years whilst the Huangshangji would have lasted more than 10 times as long. It would be interesting to see diversity versus duration of time bin plots, even if only in supplementary materials.

Reply: Thanks. Yes, the data are grouped by stratigraphic formations. This is because we report the ecosystems of these individual stratigraphic formations and do not wish to increase error by trying to generalize or combine data from different geographic regions or moderately different ages. Working on individual ecosystems from well-documented geological formations is then a strength of our approach, resolving many of the very real problems when data are combined to represent a quasi-global signal for a particular time bin.

In fact, most of the formations correlate well with independent Stages (or Ages) and show no time transgressive aspects at that level of temporal resolution, and they therefore are used as time bins. The only two exceptions are the Guodikeng and Kelamayi formations. The former saddles the P-Tr boundary and spans the Changhsingian to Induan (Early Triassic). The Guodikeng Formation also possesses many fossil horizons and yields abundant fossils throughout the entire formation. This formation therefore is separated into two units by the P-Tr boundary, representing the Changhsingian and Induan communities, respectively. The Kelamayi Formation spans the late Middle Triassic to early Late Triassic and yield abundant fossils in few horizons from its lower and upper parts, respectively. The Kelamayi Formation is also well studied, and all the literature reports fossil records from several horizons from its lower and upper parts, respectively. This formation therefore is also subdivided into two parts, and two communities represent late Middle Triassic (late Anisian-

Ladinian) and early Late Triassic (Carnian) faunas, respectively.

The lithostratigraphy and chronostratigraphy were synthesized from Yang et al (2012), Zhang (1981), Liao et al. (1987), Wartes et al. (2002), and Zhu et al. (2005). These papers studied the lithostratigraphy and chronostratigraphy from multiples sections in the northern Xinjiang basins based on plant fossils, spores and pollens, ostracods, and zircon radiometric ages from volcanic ash beds. However, the boundaries between formations may not be precisely coincident with the stage (or age) boundaries. So, we revised Fig. S1, using dash lines to mark the formation boundaries.

It is also true that stage-based biodiversity is usually biased by the different duration of various stages (or ages). This is a common problem for studies that count taxonomic numbers to represent biodiversity based on a time bin of a stage (or age). As the Reviewer points out here, both the Upper Guodikeng Fm. and Huangshangjie Fm. are two good examples, and the latter unit represents a duration almost 10 times longer than the former unit does. However, if looking at the detailed fossil horizons, we found that, although the Huangshangjie Fm. is up to 100 m thick, this formation is dominated by fossil-barren coarse sandstone/conglomerate through most of its thickness. Only about 10-m-thick (or so) strata of siltstone and mudstone yield abundant fossils, and most strata (coarse sandstone-conglomerates) do not yield fossils. The fossil-bearing strata are similar to those of the Upper Guodikeng Fm. in both lithologies and thickness. This case is particularly common in the northern Xinjiang basins. These two communities only represent the best cases in the Induan (early Early Triassic) and Norian (mid-Late Triassic), respectively, and they cannot represent the true overall or average levels of communities in both time bins (i.e. Induan and Norian, respectively).

Additionally, although the majority of studies that have demonstrated a close correlation between sampling proxies and fossil diversity have been carried out at **global or continental scales**, using sampling proxies that are arguably imprecise, such as rock outcrop area (Smith, 2001, Smith and McGowan, 2005, Smith and McGowan, 2007, Wall et al., 2009, Wall et al., 2011) and counts of fossiliferous formations (Peters and Foote, 2001, Peters and Foote, 2002, Wang and Dodson, 2006, Fröbisch, 2008, Barrett et al., 2009, Butler et al., 2009, Benson et al., 2010, Benson and Butler, 2011, Butler et al., 2011, Upchurch et al., 2011). In contrast, the majority of studies carried out at **finer geographic and stratigraphic scales** have found little evidence for a strong correlation between the rock and fossil records (Benton et al., 2004, Mander and Twitchett, 2008, Benton, 2012, Dunhill et al., 2012, Dunhill et al., 2013, 2014).

Following the reviewer's comments, we also added Diversity versus Duration of time bin plots in the supplementary materials (see Fig. S4). Diversity does not correlate with Duration of time bins (Spearman rank-order correlation, $r_s=0.41538$, $p=0.13967$) in Xinjiang.

Geological setting: there needs to be some discussion of climate changes [if any] in the Junggar Basin over this long span of time.

Reply: Thanks. We added documentation of climate changes in the studied area over the Permian to Jurassic.

122-30: this paragraph is the crux of the paper, and it is beyond my pay grade to say if the exponential - power law modelling is valid or not, I presume it is, but I'd have liked to know how the analysis incorporates missing elements of the communities (e.g. insects will not be represented in anything like their true importance). Also, how is abundance/occurrence accounted for, if at all? Some species may be extremely rare and so any links in the food webs will be trivial.

Reply: Thanks. This model has been used in multiple studies (e.g. Late Cretaceous communities in North America; P-Tr communities in Africa; Late Ordovician communities in the Cincinnati Basin etc.) and proved a powerful tool for palaeocommunity dynamic study. We agree that there must be some missing elements of the palaeocommunities. However, Roopnarine and Dineen (2018, Marine Conservation Paleobiology) have showed that despite the loss of species, guilds, and trophospecies interactions, particularly soft-bodied organisms, the overall guild diversity, structure, and modularity remained intact. Which means, it is reliable to reconstruct palaeocommunities using incomplete fossil records.

In this model, we use relative population size to account for abundance/occurrence. Relative population size is determined by a difference equation that sums energy input via consumption and energy output via predation:

$$N_i^x(t) = \frac{1}{N_i^x(0)} [\sum S_x N_j^y(t) - \sum S_k N_k^z(t)]$$

The above equation is equivalent to a Lotka-Volterra interaction equation, without an intraspecific interaction term. Each species is assumed to exist in a stable equilibrium in the unperturbed web, but perturbation, whether by the removal or addition of connected nodes, will alter the equilibrium of at

least one other node or species. The propagation of the perturbation through the web is recorded as displacement of species from equilibrium. Species states are updated sequentially and iteratively until no more state changes occur. Those changes cease when the impact of the perturbation has dissipated fully because of changes to population sizes, including reduction to zero.

In fact, what we build is a guild-level food-web, and then generate 100 stochastic species-level food-web by applying mixed exponential–power-law link distributions uniformly, which consistent with the hyperbolic distributions typical of modern food webs and other complex networks.

159: “tropic”!

Reply: Thanks, we revised that.

Fundamentally the extinction model assumes diversity is controlled by primary productivity because the extinctions are modeled by reducing productivity. Diversity does not equal productivity in the oceans, but is more clearly linked on land I guess? And how is primary productivity assessed, not by plant diversity I presume, is it just assumed?

Reply: Thanks. In fact, we aren’t assuming that diversity is controlled by primary productivity, but we use the extinction model (CEG) to simulate bottom-up extinctions by incrementally reducing primary productivity, and this model also permits top–down interactions and trophic cascades resulting from bottom-up perturbations by recalculating interaction strengths of species when some of their resources or predators go extinct.

Yes, primary productivity is assumed. In this model, the number for primary productivity does not equate to diversity of plant/algae. The number of available units of primary production was thus fixed for all meta-networks as 10 times the maximum species richness of primary consumer guilds, reflecting the thermodynamic scaling of energy transfer among trophic levels (repeating simulations at levels 20, 30 and 40 times, the maximum species richness of primary consumer guilds did not alter results qualitatively). By fixing this as a plausible constant for all the communities, we can then compare differences between the communities and refer those to other causes.

211-218: both freshwater and terrestrial communities are modelled together? They’re clearly different (look at the fundamentally different response during the KPg event) – hence they plot in different fields in figure 3, it would have been nice if they had not been lumped together. The description here talks about the two trophic space areas and yet for nearly half the time plots in figure 3 only one space can be seen – why?

Thanks. Yes, both freshwater and terrestrial communities are modelled together. They are different but not isolated, as their food-webs are linked by insects, amphibians and some carnivores (e.g. crocodile), nutrients flow across freshwater and terrestrial communities, thus it is fair to lump them together in our model, and lumping them together is a common method in palaeo-food web study (e.g. Roopnarine et al., (2007, 2015, 2018), Mitchell et al., (2012)).

The trophic space areas would shrink or expand in different communities; the size of trophic space represents the dominance of the guilds, where sometimes the invertebrates' space is too small to be seen (e.g. in JCY).

217: “no trophic space innovation was lost” – where can we see this in the plots?

Reply: Thanks. Our original statement is “No fundamental trophic space innovation or loss was

detected, at least for the local Xinjiang freshwater species dominated communities during this ~121 myr". If there is fundamental trophic space innovation in Xinjiang, we would find that a new "hot spot" appeared in the trophic space figure. And if there is fundamental trophic space lost, we would see a "hot spot" permanently disappear. However, we didn't see them, so we have no evidence showing that there is fundamental trophic space innovation or loss, at least, in the studied areas.

229: "extinction affected communities": why is the Capitanian considered extinction affected – the extinction happened in the latter part of this interval and so surely the Wuchiapingian community was the affected one? I also wonder what community they are assessing with their Capitanian dataset because it will include pre and post-extinction species.

Reply: Thanks, we deleted "extinction affected communities".

The G-L extinction has long been debated in terms of timing and pattern. This extinction was originally proposed for biotic extinction occurring at the Capitanian-Wuchiapingian boundary (Jin et al., 1994, *Palaeworld*), but was later denied by many authors; instead, the G-L extinction was calibrated to the mid-Capitanian (Wignall et al., 2009, *Science*; Bond et al., 2010, *ESR*). Recently, we confirmed the existence of the marine Capitanian-Wuchiapingian boundary extinction from South China (Huang et al., 2019, *Geology*). Thus, there were likely two phases of extinction, and they occurred in the mid- and end-Capitanian, respectively. This means that part of the Capitanian biota was affected by an extinction event in marine settings when all biota derived from the entire stage were counted. The pattern and timing of the terrestrial G-L extinction is still under debate (Retallack et al., 2006, *GSAB*; Lucas, 2009a *JAES*, *b Geol. Soc. Am. Abstr. Programs*; Bond et al., 2010, *ESR*; Rubidge et al., 2013, *Geology*; Day et al., 2015a *SAJS*, *b PRSB*; Fan et al., 2019, *Science*; Shen et al., 2020, *ESR*). The Permian terrestrial faunas are best recorded in the Beaufort Group of South Africa's Main Karoo Basin, and the latest studies suggest that the G-L crisis was a stepwise extinction in the late Capitanian and likely a global event (Day et al., 2015a *PRSB*; Lucas, 2017a *ESR*). Likely, there is an extinction phase within the Capitanian, like that in marine records.

In Xinjiang, the QZJ community represent the Capitanian assemblage, and thus comprises pre-extinction taxa and likely some during-extinction/post-extinction species. The low ecological stability of the QZJ community may suggest an earlier ecological degradation before the G-L boundary, or the stability decreased because the QZJ community mixed with some during-extinction/post-extinction elements. Instead, the relatively high stability of the Wuchiapingian community indicates a relatively fast rebound and recovery of the post-extinction communities in terrestrial ecosystems, similar to the coeval (early Wuchiapingian) marine communities in South China that comprised abundant brachiopods and indicated a fast recovery of benthic communities (i.e., Chen et al., 2005 *PPP*; Shen & Shi, 2007 *Palaeworld*).

Here we are also told the Rhaetian community was extinction affected although later (line 272) we are told this interval was "the prelude to the T-J extinction"?? – so why does the collapse threshold change before the terminal mass extinction? I suspect it is because of the enormously different durations of the Norian and Rhaetian – the former is ~5 times longer than the latter. Hence the diversity in the former could well be five times higher.

Reply: Thanks, we deleted "the prelude to the T-J extinction".

Like the Capitanian community, the Rhaetian community may also have been affected by extinction,

or this interval was ‘the prelude to the T-J extinction’. The timing and pattern of the T-J extinction has also long been disputed, in particular, the terrestrial T-J extinction. Lucas et al. (2015) argued that numerous clades experienced much lower diversity in the Rhaetian rather than focusing the entire reduction in diversity at the T-J boundary. Our new work shows that the collapse threshold indicates an overall decrease in the number of guilds and guild richnesses in the HJG community, strengthening evidence that that crisis might have occurred within the Rhaetian, although its occurrence at the end of the Rhaetian (also T-J boundary) cannot be excluded. If so, some extinction-affected taxa from the Rhaetian were also collected. Again, similar to the aftermath of the G-L, the early Jurassic communities have relatively faster recovery after the T-J extinction than the P-Tr extinction. This is probably because a low-amplitude, and earlier (within the Rhaetian rather than the end of this stage) extinction occurred before the T-J, and thus the biota took a shorter time to recovery after the T-J crisis.

In a word, as demonstrated in Fig. 5, the collapse threshold occurs before the G-L and T-J boundaries, respectively (Fig. 5). These are new observations obtained from the present study, and they indicate the difference between the P-Tr extinction and both the G-L and T-J extinctions. We discuss this issue in the text. During the P-Tr transition, the collapse threshold occurs immediately after the Changhsingian-Induan boundary (also P-Tr boundary), and the affected communities occur after the P-Tr extinction. In contrast, the affected community (also collapse threshold) occurs before the G-L and T-J extinctions, respectively. This could be because the P-Tr extinction has a more abrupt, much higher amplitude of extinction than both G-L and T-J extinctions. Thus, the post-extinction biota of the P-Tr has a much-delayed recovery than that following the G-L and T-J extinctions.

As argued above, the Norian is much longer than Rhaetian in duration. However, the Huangshanjie Fm. has similar fossil horizons to that of the HJG Fm., and both fossil-bearing strata are similar in the number of horizons, lithology and thickness. Thus, these two communities from these two formations each represent a similar time duration, although the Norian stage is much longer than the Rhaetian stage. Again, the HSJ and HJG communities represent the best cases in these two time bins, rather than representing the overall biotic levels or average levels.

Anyhow, to minimize sampling bias, we also discuss the sampling issues in the Supplementary materials, and we test the hypotheses that the observed differences between communities are the results of under-sampled communities by reducing the richness of all the communities to match those of the HJG community (the lowest species richness community) and comparing the resulting CEG dynamics with those of the observed communities. The resulting collapse thresholds were statistically indistinguishable between the two sets of simulations. In your second comment, we also show that diversity is not correlated with duration of time (Spearman rank-order correlation, $r_s=0.41538$, $p=0.13967$). To sum up, taxonomic richness/various durations alone is no predictor in the CEG model.

However, later, even the existence of the TJ mass extinction is questioned (lines 370-1). Ideas about this mass extinction seem to evolve through the manuscript – they’re certainly not consistent.

Reply: Thanks, as argued above, we re-worded some sentences to keep the consistency of our treatment of the T-J extinction in the text.

298: “Here, we discovered....” None of this was actually discovered, it was already well known before – this is a meta-analysis of previously published work after all.

Reply: Thanks, we delete “discovered”, and re-worded this sentence.

303: community size? An odd phrase, even if there were only a few species they could be present in huge numbers. “Community diversity” is better.

Reply: Thanks, we changed it to “community diversity”.

310: “revealed major disruptions” – the modeling did not reveal this because it was assumed in the modeling. Getting a handle on terrestrial biomass/productivity is not possible – the quoted references equate diversity with biomass but do not assess productivity in any way. Modeling like this can suggest/guess that productivity has collapsed – it’s a reasonable thing to model - but no more.

Reply: Thanks! These corrections/suggestions are followed in the revised version.

388: “relative magnitudes” – of what? Lava volumes?

Reply: Thanks, we change it to “relative extinction magnitudes”.

390-4: this is all a bit vague and waffly. There’s talk of extinctions with sufficient magnitude to remove guilds – which one was this? The PT extinction has just been noted to have not removed any guilds in freshwater settings?

Reply: Thanks, we re-worded this section, makes is more clearly.

We detect a significant loss of guilds during the P–Tr transition (e.g. large and very large herbivores, molluscs I (bivalve) and molluscs II (gastropods)), but this great loss is not permanent, and these empty guilds will be refilled by new species in the recovery stages. We noted that the PT extinction removed some guilds in freshwater settings (not permanently), but did not alter the original trophic space structure.

398-9: “The G–L and T–J extinctions were each preceded by low-stability communities” but it has just been noted that the G-L extinction actually happened within the Capitanian – so the “low stability” community will likely be the post-extinction community intermixed with the pre-extinction one.

Reply: Thanks! Yes, as stated above, your suggestions are followed, and we re-worded this sentence.

The so-called T-J pre-extinction community is likely just an artifact of different time bins (see above).

Reply: Thanks! As argued above, the T-J pre-extinction community may include some pre-extinction taxa and extinction-affected taxa.

The T-J pre-extinction community is not an artifact of different time bins, because taxon richness/various duration alone is no predictor in the CEG model (see above for details).

Referee: 2

Comments to the Author(s)

General comments: In this manuscript, the authors use a modeling approach (Cascading Extinctions on Graphs; CEG) to examine changes in community stability over time in the context of three major mass extinction events (the end-Guadalupian, the Permo-Triassic, and the Triassic-Jurassic). In some ways this paper is very exciting. The CEG model has been used extensively to examine Permo-Triassic tetrapod communities in southern Africa, and a now well-established narrative of the changes those

communities underwent (in the context of the model) has been built up. This paper is the first application of the approach to terrestrial Permo-Triassic communities in a very different geographic area, and it is very interesting to see some of the same patterns of change emerging. The paper also applies the model to a set of communities spanning a longer time interval and more mass extinctions than has been the case in southern Africa, and this helps to contextualize some of the results seen in both places. In particular, this longer time interval emphasizes the uniqueness to the Permo-Triassic extinction compared to the others. These are important results. However, the manuscript also has some issues that I think need to be addressed before it is ready for publication.

Reply: Thanks for the positive comments! Most suggestions and comments are followed in the revised version.

1) At a broad level, I think there is a big missed opportunity in this manuscript. The small hint in the title notwithstanding, in reading this manuscript you would think that the communities here are basically the same as those studied in southern Africa, and I bet most readers would come away thinking that the big message is that tetrapod communities in Africa and China were affected by the PTME in the same way. However, the communities here actually differ considerably from the African ones in the sense that they have a much more heavily sampled aquatic invertebrate fauna. That's potentially a very significant difference, but it hardly gets mentioned anywhere. The fact that this different kind of community (or at least differently sampled community) shows many of the same kinds of changes through the Permian and Triassic strikes me as an important point because it implies that different types of communities were affected in much the same way by the PTME or that very different sampling regimes applied to more or less the same underlying terrestrial community will recover a consistent signal of ecosystem change. In my mind, those results are at least as important as the observation that overall patterns of stability over time are similar here to what's observed in southern Africa, and they really deserve some consideration in the manuscript. Indeed, they would constitute a shift from "more of the same" to "unique insights made possible by this new dataset".

Reply: Thanks. This suggestion is very important. To address this new perspective, we add this point in the Discussion (Section 4(a)). We also emphasized this difference in the Abstract and Conclusions.

2) Something I bring up in many of my comments below is that I'm surprised that the authors don't apply the approach of Roopnarine et al. (2018) for quantifying differences in the CEG results (i.e., using a combination of intercept and Hill's slope). The approach here considers a rough equivalent of only one of those parameters, but there's relevant variation in the results that would and should be captured by the other. Indeed, the Roopnarine et al. (2018) paper is probably the most relevant to this manuscript of all the CEG papers to date, both in terms of methods and conclusions, but it gets very little attention. Beyond including a method that would strengthen many of the results, I think the general message of that paper would further bolster many of the conclusions of this one.

Reply: Thanks, we agree with you that some general messages of that paper are added in the text.

Hill's Slope used in Roopnarine et al. (2018) was effective for the Karoo communities CEG results, they used a general model: $y=b_0+b_1/[1+e^{b_2(x-b_3)}]$.

However, the CEG results in this study cannot be fit satisfactorily with a general model. For example, the JCY community's CEG results fit well with this model ($y=b_0+b_1/[1+e^{b_2(x-b_3)}]$), $b_0=0.1273964$, $b_1=0.7728444$, $b_2=26.49895$, $b_3=0.6332952$.

But, the HSJ community's CEG results do not satisfactorily fit this equation ($y=b_0+b_1/[1+e^{b_2(x-b_3)}]$), $b_0=0.0733379$, $b_1=0.8992705$, $b_2=25.55404$, $b_3=0.699046$. Here the b_0 (Parameter b_0 is the expected level of extinction when perturbation is very low) is higher than the average extinction level at low perturbation level.

We have tried multiple models, but we can't find a general model to sufficiently fit all the CEG results for the Xinjiang communities.

In an earlier version of this manuscript, we also used the PCA method (Mitchell et al., 2012, PNAS) to quantify community robustness at relatively low perturbation levels (10, 20, 30, and 40%); the PCA results showed that communities with high PC scores usually have a low collapse threshold. So, we removed the PCA sections because of limited length of a paper.

As you say, how to best quantify aspects of the CEG output for comparison across communities is a long-running issue. We think it depends on the CEG results for the studied communities, if there

is a general model to fit all CEG results of studied communities, it would be better to use the method in Roopnarine et al. (2018, JVP).

In this revised manuscript, we add the PCA results in the supplementary material, and these results are a complement to the changepoint analyses.

Beyond these issues, there's some other methodological items that I think need to be addressed (e.g., discussion of how insect richness was estimated; some additions to the rarefaction analyses), areas where some clarifications are needed, changes to figures, and a few places where I think the authors misrepresent some of their results and those of other relevant papers (not in a malicious or intentionally misleading way). These all strike me as more minor problems in terms of scope, but they definitely should be addressed.

Reply: Thanks so much. We revised all these problems based on your suggestions (see below).

Line 26 to 30: This sentence seems needlessly complex. I recommend dividing it up into at least two.

Reply: Thanks, we divide it up into two sentences.

Line 32: the lower in lower Permian should not be capitalized. Also, if word count allows, I recommend adding "the" before both lowers. Finally, change to shows (its subject, modelling, is singular).

Reply: Thanks, we double checked this manuscript, and revised all the same issues.

Lines 46-47: remove any and insert mechanisms after killing

Reply: Thanks, done.

Line 56: I guess this sort of depends on what you consider similar. Of the three, continental configuration might be the most similar, but there were some fairly substantial changes in things like atmospheric CO₂ and O₂ levels, and global temperatures over this interval.

Reply: Thanks, agree, we replace "atmospheric compositions, climates" by "large igneous province eruptions".

Lines 55-56: it would be good to have some references for the differing levels of marine vs. terrestrial diversity at the different times. Also, this is kind of a run-on sentence. I recommend splitting it up into two where the colon currently is.

Reply: Thanks, here we use terrestrial tetrapod genera extinction rate to replace the plant extinction rate and split it into two sentences.

Line 74: "trophic food web" seems a little redundant

Reply: Thanks, we delete "trophic".

Line 77: change to graphs

Reply: Thanks, done.

Line 90: change from to during

Reply: Thanks, done.

Line 92: examples of biostratigraphic and geochronology refs would be good to add here

Reply: Thanks, we add a few refs here.

Line 102: change were to are. Also, a more general comment about the methods. Only a couple of your communities have insects actually preserved in them (Table S1), but I see you have insects as part of all of your food webs (Table S2). The richnesses of the guilds vary across a fairly narrow range, so I assume you are estimating insect richness somehow (perhaps following the method used in previous CEG work in on African and North American paleocommunities). You should state that you are doing this, and note how you are making the estimates.

Reply: Thanks! Yes, you are right; we use the method in Mitchell et al. (2013, PNAS) to estimate the guild richness of insects. We add a section to explain how we are making the estimates.

Line 110: I recommend using metanetwork instead of matrix for consistency with previous work using the CEG framework

Reply: Thanks, “food-web matrix” is replaced by “food-web metanetwork” in this manuscript.

Line 111: In previous works applying CEG to Permo-Triassic tetrapod communities, some basic rule were applied to structure interactions between guilds. Based on Fig. 1, it looks like those rules, or a slightly modified version, were applied here as well. I think it would be good to note that somewhere, either here in the methods or in the supplement.

Reply: Thanks! Yes, in this paper, we apply the same rules as in Roopnarine et al. (2007, 2015) and Mitchell et al. (2012) to structure interactions between guilds. We add a few sentences in the Method section to note these rules.

line 112: change to early Permian

Reply: Thanks, done.

Line 130: If possible, include a couple citations to help support the observation that species level network variation doesn't strongly affect the results in most cases.

Reply: Thanks, here we add three references to support this observation.

Lines 133-138: Is this actually in the right place? It seems to be presenting results, and presenting them before you actually describe the methods being used.

Reply: Thanks, you are right. We rewrite these sentences, describing the NMDS method.

Line 149: change to prey. Also here, and in line 152, I recommend using metanetwork for the guild-level food web, similar to my earlier comment.

Reply: Thanks, done. We replace each “guild-level food-web” by “metanetwork” in this manuscript.

Line 160: you seem to be missing a parenthesis

Reply: Thanks, done.

Line 160-162: I kind of disagree with this statement. CEG doesn't directly test for variation in

structural composition or guild richnesses. Those are input data for the model inasmuch as communities being examined in a set of analyses vary in those properties. Instead, it models the effects that those differences have on the response to the applied perturbations. The NMDS analyses you describe above, and the network properties to measure provide a further quantitative description of these differences, but again don't really represent a test in the sense of trying to falsify a hypothesis. I think a better way to describe things is to say that your combined approach (i.e., NMDS, network metrics, CEG results) allow you to describe how the structures of the communities differ, and then determine how those differences translate into community performance in the face of perturbations. The latter part of this could be construed as a test of whether the described differences have a significant effect on community performance.

Reply: Thanks, we agree, the CEG can calculate the guild richnesses but not test guild richnesses. We revised these sentences based on your suggestion.

Line 165: change was to were

Reply: Thanks, done.

Line 180: One of the long-running issues of work in the CEG framework has been how to best quantify aspects of the model output for comparison across communities. Change point analysis is one way to do this, but I also thought that the combination of looking at intercept and Hill's Slope used in Roopnarine et al. 2018 was effective. Indeed, it captures an aspect of performance (intercept) that is not considered in the current analysis, but that shows considerable variation in the southern African Permo-Triassic communities in the Roopnarine et al. paper. Is there a reason why that approach wasn't used here? If possible, I would suggest using it at least as a complement to the change point analyses.

Reply: Thanks! We explained why we can't use Roopnarine et al. (2018)'s method, so we use PCA to capture the CEG dynamics at relatively low perturbation level (following Mitchell et al., 2012), and we add the PCA results in the supplementary material.

Fig. 1: I recommend putting quotes around *Chasmatosuchus*; that seems to be current practice in the archosauromorph literature. Also I think the photo of the specimen will be a bit harder for non-specialists to read than the others, since the face isn't as obvious.

Reply: Thanks, we add quotes around the taxa names in Fig.1.

Fig. 2: The color scheme used definitely needs to be changed to make sure it is accessible to readers with colorblindness.

Reply: Thanks, this is a very considerate suggestion! We changed the color scheme in this figure.

Line 213: I think you mean to call out fig S2 here.

Reply: Thanks, we delete this note.

Line 217: I recommend noting, maybe parenthetically, that such innovation would be manifested on the plots in Fig. 3 as the rise of a new peak on the landscape.

Reply: Thanks, agree, we add a few words to note readers.

Fig. 3: Here again, I think you should avoid using a color ramp that uses red-green differences so strongly.

Reply: Thanks, we changed the colour scheme in this figure.

Line 230: change to periods; change to late Permian

Reply: Thanks, done.

Line 232-233: I think it would be better to say that diversity is correlated with connectance (i.e., omit the “changes”)

Reply: Thanks, done.

Line 243: This variation in response to low perturbations isn't really captured in Fig. 5, but in some ways it's one of the most interesting parallels to previous results from southern Africa. This again makes me think using the intercept/Hill's slope approach used by Roopnarine et al. (2018), with a bivariate plot showing communities in that space, would provide a fuller description of the differences among communities.

Reply: Thanks, yes this isn't really captured in Fig. 5, we use PCA method to capture the CEG dynamics in response to low perturbations, we add PCA results in supplement material.

Line 289: change did to does

Reply: Thanks, done.

line 294: change functionally to functional

Reply: Thanks, done.

Line 310-313: I disagree with these statements. The approach you're using here cannot reveal anything directly about changes in primary productivity because the units of productivity used in the model are estimates based on herbivore diversity (i.e., they are not actual observations about the floras at the different times). At best they can tell you that the consumer communities at a given time might have required more or less primary productivity to be available than at some other time, but that is a very different statement. It only sets a minimum floor on required productivity, but it doesn't say anything about what the actual available productivity was. Likewise, it was never claimed in any of the papers on southern African Permo-Triassic communities that actual productivity levels declined. A more accurate thing to say here would be that the observed changes in animal diversity suggest a decline in the amount of productivity that is needed to support the communities, and that in turn is perhaps consistent with the fact that paleobotanical evidence suggests that suggests a drop in plant diversity at this time. You can also note that a similar implied drop in productivity requires was observed in the African communities.

Reply: Thanks, agree. Reviewer 1 also pointed out this issue. We delete “revealed”, “suggest/guess/assume” are more appropriate words to use, we also revised these sentences based on your suggestions.

Line 317: This is kind of a weak statement. I think you could strengthen it by talking a bit more about the potential implications of the diversity decline. I understand that to be a change from a diverse,

more forested environment to a more open habitat dominated by a few species of plants such as *Pleuromaia*. Describing that transition in a little more detail would help to make the expectation of reduced productivity more obvious.

Reply: Thanks, here we add a little more detail about that transition and add a few references.

Line 322: I think some readers would find the inability to localize the cause of the change to one or more of the variables unsatisfying because they are unfamiliar with the non-linear dynamics of the CEG model, which in turn make such predictions quite difficult. I think noting this difficulty will help to reduce that concern and explain why there isn't an obvious factor you can point to as the cause.

Reply: Thanks, we add a few words to explain that.

Line 330: I recommend moving the citation for ref 38 to the end of this sentence.

Reply: Thanks, done.

Line 337: I know it's a drum I'm beating a lot in this review, but considering intercept here would be good as well. It too has returned to "normal" in the CEG plots by the Middle Triassic.

Reply: Thanks, agree. So, we add PCA results in supplement material.

Line 339: I would expand this to Southern Africa more generally, following the results of Roopnarine et al. (2018), and recommend citing that paper here. In many ways, it is methodologically the most directly comparable of all the CEG work to your manuscript, and it helps to further establish the apparently broad geographic pattern of recovery in the Middle Triassic.

Reply: Thanks, agree, we cite Roopnarine et al. (2018) here. We also add more messages from that that paper and compare the results in Karoo and Xinjiang.

Line 340-345: I think there's a couple ideas here that should be unpacked a little more for clarity. 1) High connectivity, or the emergence of a high connectivity guild, might help to stabilize the community in the model because of changes in network dynamics (and perhaps regardless of whether the guild was itself a generalist or a specialist that happened to be preyed on by many other guilds); 2) generalist trophic habits for the guild that would tend to make it less likely to be fatally impacted when the community is disturbed. It seems like your implying that the notostracans were generalists (and thus had lots of food sources they could draw upon when the community was disturbed) and that they were preyed upon by a lot of other guilds (and thus were a reliable food source for other taxa), so a bit of both 1 and 2. Do you think that same combination applied to the other highly connected guilds you mention?

Reply: Thanks, we unpacked these ideas a little more for clarity in this revised manuscript. We don't think the same combination could apply to the other highly connected guilds we mention, because as you say, a high connectivity guild doesn't mean it consists of generalists; specialists that happened to be preyed on by many other guilds could also make up a high connectivity guild.

Line 377: That's not exactly how I would characterize the message of the Mitchell et al. paper. Instead, their result was more that the Maastrichtian communities in particular were less resistant on average than the Campanian ones. It is that lower resistance of the Maastrichtian communities that has implications for the K-Pg extinction (i.e., it might not have been quite as bad if the asteroid hit in

the Campanian...).

Reply: Thanks, we have read Mitchell et al. (2012) many times, and that paper suggested that Maastrichtian communities were less stable than Campanian communities, and a weak Maastrichtian community may have exacerbated the impact of the end-Cretaceous extinction. Our original expression may lead to some misunderstanding, so we revised this sentence based on your comments.

Line 397: Change to: "Our results indicate three significant decreases in paleocommunity stability in Xinjiang, coinciding with the G-L, P-Tr, and T-J extinctions."

Reply: Thanks, done.

Line 388: I'm not sure how much space is available to do it, but I think this is an idea that should be developed further. The pattern of decreasing stability leading up to an extinction vs. following an extinction seems significant, and I have a hard time thinking that just the size of the event is the only thing at play (as opposed to the rate of environmental changes, details of environmental effects, etc.). You might be right that it's just an issue of magnitude, but you should present more evidence to help support this claim.

Reply: Thanks! Yes, the different CEG dynamics among these extinctions are very interesting, magnitude is just one of the explanations, we can't exclude other explanations (such as the rate of environmental changes, details of environmental effects, etc.). We add more messages to state this issue.

Line 403: insert that after confirmed

Reply: Thanks, done.

Line 438: change to Botha-Brink

Reply: Thanks, done.

Supplement, Sampling issues paragraph 1: I think you mean to call out figure S3 here.

Reply: Thanks, done.

Supplement, Sampling issues paragraph 3, line 2: change to communities.

Reply: Thanks, done.

Supplement, Sampling issues general comment: This approach refutes the idea that sampling is an issue if the sampling probability is more or less the same across all guilds and across all time intervals. However, looking at table S1, there are some patterns that make me think this is definitely not the case (e.g., insects only being preserved in the Late Triassic, the near complete absence of bivalves in the Early Triassic, despite their considerable diversity at other times). Therefore, in addition to your random re-sampling, it might be good to do some more targeted manipulations to see what effect non-random patterns of sampling might have (e.g., what are your Permian communities like if there are no or very few bivalves in them?). These sorts of experiments might also be helpful in nailing down some more specific effects of particular guilds, such as the highly connected ones you mention in the main text.

Reply: Thanks! The purpose of this approach is to see whether undersampled taxon richness caused the results seen for the Xinjiang communities. To do some targeted manipulations is a very helpful suggestion, and we add more experiments to see if the highly connected Notostraca would help to stabilize the community and if bivalves made the differences in CEG dynamics between Permian and Triassic communities.

Below are the HSJ (original) and HSJ (reduced 90% of Notostraca) CEG plots. It is very clear that the community robustness is significantly reduced at low perturbation level, and the collapse threshold decreased, suggesting that the Notostraca guild will help to increase in community stability.

Below are the LCG (original), LCG (reduced 90% of Bivalve), HYC (original) and HYC (reduced 90% of Bivalve) CEG plots. The community robustness is slightly reduced at low perturbation level in both LCG (reduced 90% of Bivalve) and HYC (reduced 90% of Bivalve), and the collapse threshold is decreased in LCG (reduced 90% of Bivalve) but no significantly change in HYC (reduced 90% of Bivalve). The pattern of LCG (reduced 90% of Bivalve) and HYC (reduced 90% of Bivalve) resistance appears to be transitional between Permian and Triassic communities, thus, we think that to some extent, the changes in bivalve guilds made the difference between Permian and Triassic local communities in Xinjiang.

Fig. 54

Referee: 3

Comments to the Author(s)

This study analyzes the changes in terrestrial and freshwater communities across three mass extinctions, namely the end-Guadalupian, end-Permian and end-Triassic mass extinctions. The authors use data from northern Xinjiang, North West China to assess changes in species diversity, guilds and trophic space. This is a comprehensive, well-documented paper with globally relevant results. The most significant result was confirming the uniqueness and extent of the end-Permian mass extinction compared to the other two crises, although they also found some generalities in all three events, which is also interesting. This is a well-written, well-explained documentation of new data that makes for a significant contribution towards mass extinction studies, and thus, I recommend publication as is. I picked up a few very minor typos in the text, which I attach as a separate document.

Reply: Thanks for the positive comments! All points of your comments listed on the attached file are followed in our revised version of this manuscript.

Reviewer(s)' Comments to Author:

Referee: 4

Comments to the Author(s).

This paper represents a thorough examination of ecosystem structure through three mass extinctions in a single area. The paper had already been through a round of reviews before it came to me. I read through the manuscript first but also had a quick look at the response to the previous reviews. On the whole I think this is an interesting paper which provides a new perspective not only on the specific ecosystems in question but also on mass extinctions in general. I think the authors have thoroughly responded to the reviewers' previous concerns and have produced an excellent contribution. I have just a couple of method questions that would be good to clarify and then I then I think it would be ready to go.

Where you are using the method of Mitchell et al. to estimate insect richness, how do you then divide insects between predators, herbivores and omnivores? At random? Based on observed insect faunas?

Reply: Thanks. Total insect richness per community was calculated on the basis of the method outlined in Mitchell et al. Total richness was then partitioned among herbivorous, omnivorous and carnivorous insect guilds using the ratios observed in insect-rich paleocommunities from the upper Permian *Daptocephalus* Assemblage Zone and Middle Triassic *Cynognathus* Assemblage Zone of the Karoo Basin in South Africa. We deemed those communities to be suitable analogues given their chronological and taxonomic overlaps with our Xinjiang communities. We add a few words to clarify this issue in the revised manuscript.

Skull length: were species with no skull left out or was their size category estimated?

Reply: Yes, when species have no skull left, we estimated their size category (mostly based on femur and vertebrae).

Perhaps in the supplement indicate which diet and size category you are assigning the vertebrate taxa?

Reply: Thanks! We add a column in Table S1, indicating which guilds are assigning the taxa.

Neil Brocklehurst

Referee: 2

Comments to the Author(s).

General comments: This is the second time I've reviewed this manuscript, and I think the authors have done a good job overall in responding to my suggestions and those of the other reviewers. I have a few suggested edits below, but they're mostly pretty minor that should be easy to fix. My only remaining sticking point deals with the comment in line 349 about the modeling results suggesting a disruption of primary productivity. I still don't think that's the case because that type of disruption is assumed by the modeled scenario in the first place (i.e., the model results are generated by removing producers). The authors note several other lines of evidence to suggest that such a disruption of producers occurred in Xinjiang near the PTB, which is good, but those are the data sources that need to be cited for that inference. All the CEG model can tell you is what the potential effects of such disturbances to the community were, not whether the disturbance itself happened.

Reply: Agree, CEG models do not suggest disruptions in primary productivity, line 349 is kind of misleading, we revised line 349 based on your suggestion. Many thanks!

Line 31: change to: and is flanked

Reply: Thanks, done.

line 80: I think 'here' can be removed

Reply: Thanks, 'here' is removed.

line 112: early (and middle and late) shouldn't be capitalized for the Permian (double-check throughout the manuscript that this is consistent

Reply: Thanks, done. We double-check throughout the manuscript (and the electronic supplementary material), make sure early (and middle and late) don't capitalized for the Permian.

line 112: delete the before northern

Reply: Thanks, done.

line 116: delete the before north. Also, I don't think north needs to be capitalized here.

Reply: Thanks, done.

Line 122: change were to are

Reply: Thanks, done.

line 136: change while to whereas

Reply: Thanks, done.

Line 145: I think this paragraph (lines 145–154) would actually fit better in the previous section (i.e., Database). Its current placement sort of interrupts the flow of information your providing about assembling the food webs.

Reply: Thanks, agree. We move this section to the previous section (Database section).

line 146: I don't think you need to capitalize North

Reply: Thanks, done.

line 178: Would a better title for this section be something like "Modelling of Palaeocommunity Dynamics"?

Reply: Thanks, agree. We change the title to "Modelling of Palaeocommunity Dynamics".

Line 202: I'm not convinced that this paragraph (line 202–223) actually needs to be a distinct section. I think it would be better to incorporate it into the previous section, probably towards the middle of that section. That ways the logical progression would be 1) introduction to CEG (~ lines 179–194 in the current manuscript); 2) details of the application of the model and quantification of results (current section f); 3) comments about the combine approach (~lines 195–200 in the current version). There's a little bit of repetition between the two sections that would need to be dealt with, but that should be pretty simple to do.

Reply: Thanks! We re-arranged these sections based on your suggestion.

Fig. 1: I think you misunderstood my previous comment here. You don't need quotes around all the taxon names in the photos, just for "Chasmatosaurus" (e.g., there's no taxonomic issues for *Urumchia* that I am aware of that would necessitate it being in quotes).

Reply: Thanks, we delete the quotes around *Urumchia*.

Fig 4. Check the color scheme here for colorblindness accessibility. The red/green combination likely should be changed. Can you switch to the color palette used in Fig. 3 (which looks nice!).

Reply: Thanks, we switch it to the color palette used in Fig. 3.

Lines 337–340: This sentence seems kind of awkward. I think you should at least change the comma after amphibian to a semi-colon.

Reply: Thanks, done.

Line 349: I still disagree with this statement. The CEG models do not suggest (any more than they reveal) major disruptions in primary productivity. They only indicate that IF there were major disruptions in productivity, they likely would have resulted in increased sensitivities and decreased resistance. Whether such disruptions in productivity occurred (i.e., whether the scenario you are modeling is accurate) must be determined from other data, not the model. Similarly, the results for the Karoo do not imply a drop in productivity, they assume that scenario in the model. In the case of the Karoo, that scenario was chosen because at the time of writing, the prevailing idea for environmental change in the Karoo at the time included things like large plant die-offs, increased erosion, etc. This may seem like semantics, but to me there is a very fundamental distinction between what is model input vs. output. A way to deal with this might be to restructure this paragraph so that your comments about the paleobotanical record and the drop in animal diversity come first, and then present the (modeled) implications.

Reply: We agree with that, the modeling results can't suggest a disruption of primary productivity. Line 349 is kind of misleading. We restructured this paragraph based on your suggestion. Many thanks!

Line 384: I think you should note what the conclusions of those studies have been. That will provide more useful context for the statements you make in the rest of the paragraph.

Reply: Thanks, we add a few words to note the conclusions of those studies.

Line 434: I'm not sure depleted is the right word here. Would "much lower threshold values" be better?

Reply: Thanks, done. Yes, "much lower threshold values" is much better.